# Conformational trajectory of allosteric gating of the human cone photoreceptor cyclic nucleotide-gated channel

Zhengshan Hu[1], Xiangdong Zheng[1] & Jian Yang [1] ✉

Cyclic nucleotide-gated (CNG) channels transduce chemical signals into electrical signals in sensory receptors and neurons. They are activated by cGMP or cAMP, which bind to the cyclic nucleotide-binding domain (CNBD) to open a gate located 50-60 Å away in the central cavity. Structures of closed and open vertebrate CNG channels have been solved, but the conformational landscape of this allosteric gating remains to be elucidated and enriched. Here, we report structures of the cGMP-activated human cone photoreceptor CNGA3/CNGB3 channel in closed, intermediate, pre-open and open states in detergent or lipid nanodisc, all with fully bound cGMP. The pre-open and open states are obtained only in the lipid nanodisc, suggesting a critical role of lipids in tuning the energetic landscape of CNGA3/CNGB3 activation. The different states exhibit subunit-unique, incremental and distinct conformational rearrangements that originate in the CNBD, propagate through the gating ring to the transmembrane domain, and gradually open the S6 cavity gate. Our work illustrates a spatial conformational-change wave of allosteric gating of a vertebrate CNG channel by its natural ligand and provides an expanded framework for studying CNG properties and channelopathy.

Ion channels open and close in response to membrane potential changes or ligand binding/unbinding. Kinetic studies, both experimental and computational, show that this gating event is usually a multistep process that starts from a resting (closed) state, transitions through multiple intermediate states, culminates in an activated (open) state, and in many cases, terminates in an inactivated or desensitized state. Determination of structures of all these states is instrumental in fully understanding the molecular mechanisms of channel gating and designing molecular tools to interrogate, manipulate and utilize gating intermediates in physiological and pathological conditions. Numerous structures of virtually all known ion channel types in various conditions have been solved, and great strides have been made recently in obtaining structures of ion channels in different functional states, including intermediate states[1–8]. However, it remains elusive and challenging to obtain a full spectrum of structures of an ion channel in various gating states under one and the same condition and with its natural activator. In our continuing efforts to elucidate the

molecular mechanisms of gating of vertebrate cyclic nucleotide-gated (CNG) channels, we have in this study captured structures of the human cone photoreceptor CNG channel in closed, transition, pre-open and open states in the presence of its natural activator, cGMP.

CNG channels are present in prokaryotes and eukaryotes and have important physiological functions[9–11]. They play an essential role in vision and smell in vertebrates. In retinal rod and cone photoreceptors and olfactory sensory neurons, CNG channels transduce light- or odorant-induced changes in the intracellular concentrations of cyclic guanosine monophosphate (cGMP) or cyclic adenosine monophosphate (cAMP) into electrical and calcium signals[9–12]. Numerous hereditary mutations in both rod and cone photoreceptor CNG channels have been associated with degenerative visual disorders such as retinitis pigmentosa and achromatopsia[9,11,13]. CNG channels are calcium-permeable nonselective cation channels[9–11,14–18]. Although belonging to the voltage-gated ion channel superfamily, vertebrate CNG channels are largely insensitive to membrane voltage and are activated by

[1]Department of Biological Sciences, Columbia University, New York, NY 10027, USA. ✉e-mail: jy160@columbia.edu

intracellular cGMP or cAMP[9–11,16,17]. CNG channels are formed by four identical or different subunits. Each subunit contains six transmembrane (TM) segments (S1-S6), a pore-loop between S5 and S6, a cytoplasmic C-linker immediately following S6, and a cyclic nucleotide-binding domain (CNBD) directly connected to the C-linker[9–11]. The C-linker contains six α helices, named A′-F′, of which, helices A′B′C′D′ form the gating ring (we refer to this region collectively as C-linker/gating ring in some instances). The A′B′ helices of one subunit interact with the C′D′ helices of its clockwise adjacent subunit in a so-called "elbow-on-shoulder" fashion[19]. The CNBD contains a well conserved cyclic nucleotide-binding pocket, and its C-helix undergoes particularly dramatic conformational changes upon cyclic nucleotide binding/unbinding[19–24].

Structures of a number of prokaryotic and eukaryotic CNG channels have been determined in recent years[7,25–35]. These structures reveal the architectures of CNG channels and the conformational differences between closed and open states, shedding light on the molecular rearrangements underlying CNG channel gating. In simplified terms, cGMP or cAMP binds to the CNBD and causes it to move toward the membrane; this causes the C-linker/gating ring to move closer to the membrane and the N-terminal (i.e., S6-connecting) end of helix A′ to rotate and expand radially; this leads to dilation and rotation of S6 and opening of the gate. In eukaryotic CNG channels, the main activation gate is located in the central cavity and is formed by two hydrophobic amino acids in S6 (the cavity gate)[26,28–30,33,34]. Some vertebrate rod and cone photoreceptor CNG channels have an additional gate at the cytoplasmic end of the pore, formed by an S6 arginine in one of the four subunits (the arginine gate)[30,31,33,34]. This extra gate partially controls ion conduction[33].

Most CNG channel structures are in either the open state or closed state. Recently, structures of three different open states and a pre-open state have been obtained for two mutant forms of a bacterial CNG channel SthK (Y26F and R120A), offering insights into the conformational transitions involved in SthK gating[7]. Two open-state structures have been obtained for the human rod photoreceptor CNGA1/CNGB1 channel[33]. However, a detailed conformational landscape of cyclic nucleotide activation of vertebrate CNG channels remains to be elucidated.

In this work, we have obtained a total of eight structures of closed, open, and multiple intermediate states of cGMP-bound human cone photoreceptor CNG channel in detergent and lipid nanodisc. These structures provide an unprecedented view of the conformational trajectory of CNG channel activation. The distribution of different gating states in detergent and lipid nanodisc and the observation that the open state is obtained only in the latter suggest that lipids play a critical role in tuning the energetic landscape of CNG channel activation.

## Results

### Structures of cGMP-bound A3/B3 in GDN

We previously obtained an apo closed-state cryo-EM structure of the full-length human cone photoreceptor CNG channel in the detergent glycol-diosgenin (GDN)[34]. The structure shows that this CNG channel is formed by three CNGA3 (A3) subunits and one CNGB3 (B3) subunit, whose S6 contains the gate-forming arginine. In this study, we first set out to solve the open-state structure of full-length A3/B3 in GDN in the presence of a saturating concentration (2–4 mM) of cGMP. With multiple rounds of data processing including in particular the use of the 3D variability analysis tool[36] in cryoSPARC[37], we obtained three single-particle reconstructions with 74,921 (28.7%), 131,492 (50.4%), and 54,256 (20.8%) particles, respectively (Supplementary Figs. 1 and 2). They produced three different structures that likely represent three distinct states, which were named GDN_cGMP_closed, GDN_cGMP_intermediate1, and GDN_cGMP_intermediate2 (Table 1 and Supplementary Figs. 1 and 2). The resolution is 3.51 Å, 3.63 Å, and 3.61 Å,

respectively, with imposed C1 symmetry. All three structures show a heteromeric tetramer consisting of three A3 and one B3. In reference to its position relative to B3, the A3 subunits are named CNGA3$_L$ or A3$_L$ (left), CNGA3$_D$ or A3$_D$ (diagonal), and CNGA3$_R$ or A3$_R$ (right), respectively (Fig. 1a, inset), following a nomenclature used for the CNGA1/CNGB1 complex[33]. In the final structures, we were able to model amino acids A158-I159 to D609-E614, I159 to D609-L611, and A158-I159 to M607-L614 of the three A3 subunits and L206 to R643 of the B3 subunit. The remaining residues in the N- and C-termini were not modeled because of weak or missing densities.

The structure of GDN_cGMP_closed is virtually identical to that of apo A3/B3 in GDN (GDN_apo), as exemplified by A3$_L$ (Fig. 1a), with a root-mean-square deviation (r.m.s.d.) of 0.49 Å, indicating that GDN_cGMP_closed is indeed closed and has not undergone any of the gating-related conformational rearrangements. Strikingly, cGMP is

**Table 1 | Cryo-EM data collection, refinement, and validation of GDN samples**

| | GDN_closed (EMDB-28595) (PDB 8ETP) | GDN_interm1 (EMDB-28603) (PDB 8EU3) | GDN_interm2 (EMDB-28611) (PDB 8EUC) |
|---|---|---|---|
| **Data collection and processing** | | | |
| Magnification | 81,000 | 81,000 | 81,000 |
| Voltage (kV) | 300 | 300 | 300 |
| Electron exposure (e⁻/Å²) | 55.03 | 55.03 | 55.03 |
| Defocus range (μm) | −0.9 to −1.9 | −0.9 to −1.9 | −0.9 to −1.9 |
| Pixel size (Å) | 1.058 | 1.058 | 1.058 |
| Symmetry imposed | C1 | C1 | C1 |
| Initial particle images (no.) | 11,977,477 | 11,977,477 | 11,977,477 |
| Final particle images (no.) | 74,921 | 131,492 | 54,256 |
| Map resolution (Å) | 3.51 | 3.63 | 3.61 |
| FSC threshold | 0.143 | 0.143 | 0.143 |
| Map resolution range (Å) | 3.04–7.24 | 3.03–7.46 | 3.10–8.33 |
| **Refinement** | | | |
| Initial model used (PDB code) | 7RHS | 7RHS | 7RHS |
| Model resolution (Å) | 3.54 | 3.64 | 3.65 |
| FSC threshold | 0.143 | 0.143 | 0.143 |
| Model resolution range (Å) | n/a | n/a | n/a |
| Map sharpening B factor (Å²) | −123.2 | −148.4 | −122.7 |
| Model composition | | | |
| Non-hydrogen atoms | 14,809 | 14,596 | 14,560 |
| Protein residues | 1799 | 1770 | 1764 |
| Ligands | 4 | 4 | 4 |
| B factors (Å²) | | | |
| Protein | 68.71 | 73.81 | 73.87 |
| Ligand | 57.19 | 58.24 | 56.93 |
| R.m.s. deviations | | | |
| Bond lengths (Å) | 0.006 | 0.006 | 0.006 |
| Bond angles (°) | 1.14 | 0.835 | 1.167 |
| Validation | | | |
| MolProbity score | 1.39 | 1.66 | 1.76 |
| Clashscore | 4.91 | 7.03 | 9.58 |
| Poor rotamers (%) | 0 | 0.19 | 0.25 |
| Ramachandran plot | | | |
| Favored (%) | 97.26 | 96.01 | 96.28 |
| Allowed (%) | 2.74 | 3.99 | 3.72 |
| Disallowed (%) | 0 | 0 | 0 |

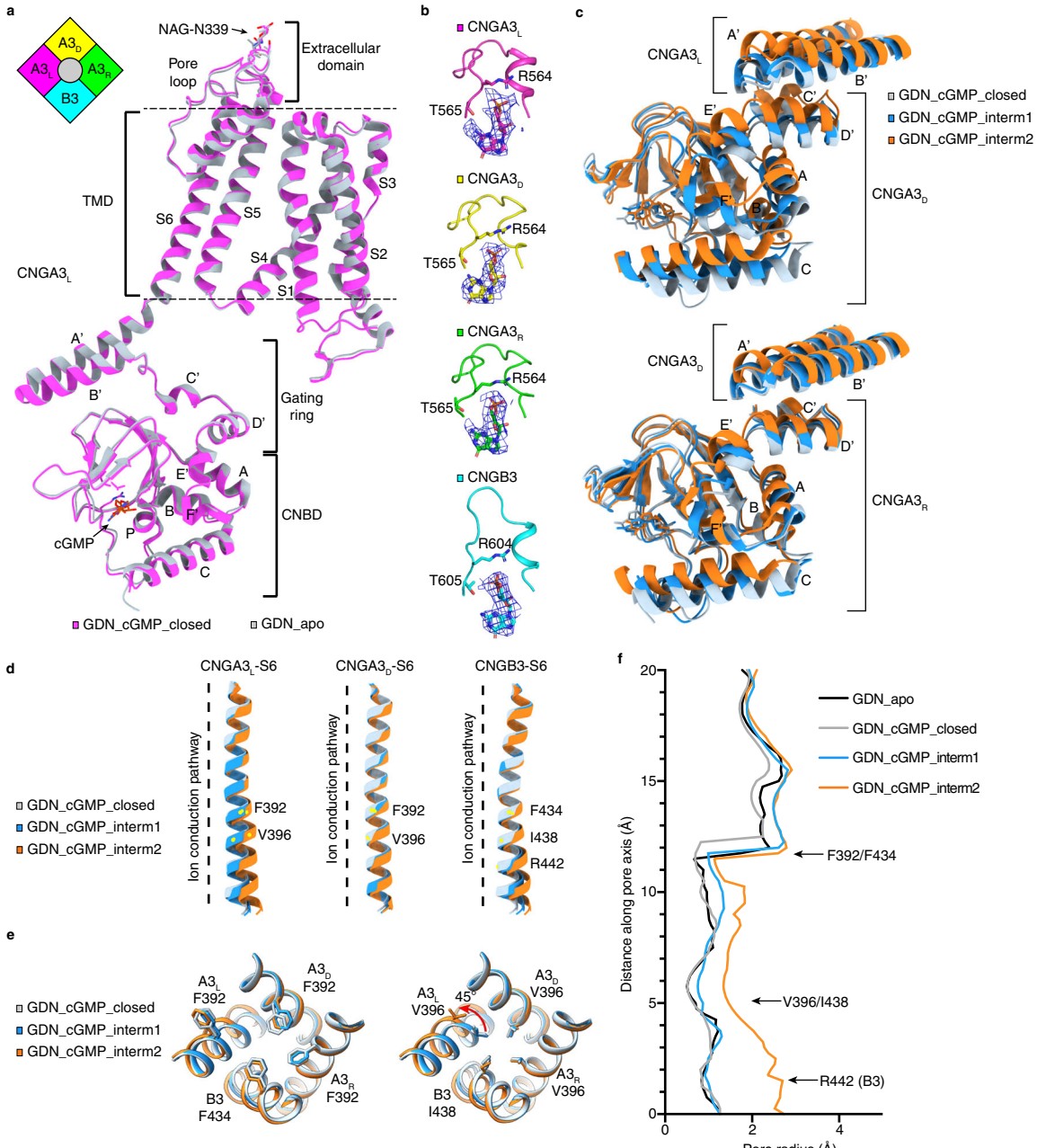

**Fig. 1 | Cryo-EM structures of cGMP-bound CNGA3/CNGB3 in detergent.**
**a** Superposition of CNGA3$_L$ protomoer of GDN_apo and GDN_cGMP_closed. Inset is a schematic subunit arrangement of CNGA3/CNGB3 viewed from the extracellular side. **b** Close-up view of cGMP in all four subunits of GDN_cGMP_closed. Cryo-EM densities are shown as blue mesh and contoured at 5σ. **c–e** Superposition of key regions of selected subunits of GDN_cGMP_closed, GDN_cGMP_interm1, and GDN_cGMP_interm2. The regions include the CNBD and C-linker/gating ring (**c**), S6 (**d**), and the cavity gate (top-down view) (**e**). The Cα positions of F392 and V396 of CNGA3 and F434, I438, and R442 of CNGB3 are shown as yellow dots in (**d**). F392 of CNGA3$_L$ has a pore projection and a sideway projection but only the sideway projection is shown in (**e**). **f** Pore-size profiles of the solvent-accessible pathway of apo and cGMP-bound states of CNGA3/CNGB3 in GDN. The origin of the pore axis is set at the intracellular end of S6, just below R442 of CNGB3.

bound to all four subunits, as illustrated by the robust cGMP densities in the well-characterized binding pockets in the CNBD (Fig. 1b). GDN_cGMP_closed, GDN_cGMP_intermediate1 and GDN_cGMP_intermediate2 show structural differences mostly in the CNBD and C-linker/gating ring (Fig. 1c and Supplementary Movie 1). As the channel advances from the closed state to the intermediate states, the C-helix of the CNBD moves up closer and closer to cGMP (which most likely results in a tighter and tighter binding of cGMP), and the A-helix of the CNBD and the E′-helix of the C-linker move up more and more toward the membrane, thereby pushing the gating ring (especially its A′B′ helices) closer and closer toward the membrane. As a consequence,

the cytoplasmic end of S6, which is connected to helix A′ of the gating ring via a mere two-amino acid linker, moves further and further away from the pore and undergoes a slight counterclockwise rotation (viewed from the extracellular side) (Fig. 1d, e).

The magnitude of the local conformational changes varies from subunit to subunit and is larger in GDN_cGMP_intermediate2 than in GDN_cGMP_intermediate1 (Fig. 1c–e). The A′B′ helices of A3$_L$ show the largest upward movement, causing S6 of A3$_L$ to undergo the largest dilation and rotation, especially toward the C-terminal end (Fig. 1c–e). As a result, the gate residue V396 of A3$_L$ rotates away from the pore in GDN_cGMP_intermediate2 but not in GDN_cGMP_intermediate1

(Fig. 1e). The other gate residue F392 of A3$_L$ has two projections (sideway projection and pore projection) with a similar occupancy in all three states (Supplementary Fig. 3a–c). In contrast, F392 of A3$_D$ and A3$_R$ and F434 of B3 have only pore projection (Supplementary Fig. 3a–c). V396 of A3 and I438 of B3 also have only pore projection (Supplementary Fig. 4a–c). R442 of B3, which projects to the pore and forms the arginine gate below the cavity gate[34], also has two orientations (up and down) along the ion conduction pathway (Supplementary Fig. 5). The up orientation is more abundant in GDN_cGMP_closed and the down orientation is more abundant in GDN_cGMP_intermediate1, while the density of R442's distal side-chain is not well resolved in GDN_cGMP_intermediate2 (Supplementary Fig. 5). These results suggest that the side-chains of F392 of A3$_L$ and R442 of B3 exhibit some intrinsic flexibility.

Due to S6 dilation and rotation, the intracellular end of the pore is slightly widened in GDN_cGMP_intermediate2 (Fig. 1f), but because S6 dilation and rotation occur mainly in A3$_L$ and are limited in magnitude, the hydrophobic cavity gate collectively formed by F392 (A3)/F434 (B3) and V396 (A3)/I438 (B3) is still tightly closed in both intermediate states (Fig. 1e, f). These results show that although cGMP is fully bound to A3/B3 in GDN and produces conformational changes in the CNBD, the gating ring, and even S6 of some subunits, these changes are not large, strong and widespread enough to open the pore.

## Structures of cGMP-bound A3/B3 in POPG/POPC nanodiscs

Many possible reasons could account for why cGMP-bound A3/B3 is closed in GDN. We considered three particular possibilities: (1) GDN binds to A3/B3 and inhibits pore opening. (2) Some copurified native lipids bind to A3/B3 and inhibit pore opening. (3) Certain lipids are needed for A3/B3 to open but are missing. To test these non-mutually exclusive possibilities, we attempted to reconstitute full-length A3/B3 in nanodiscs with soybean lipids, which we used to solve the cGMP-unbound closed-state structure of TAX-4[28]. However, our effort failed repeatedly due to various technical reasons, including protein precipitation and poor-quality single particles. We took three approaches to solve these problems. First, to increase A3 and B3 expression level and protein stability, we removed amino acids 1–151 and 1–78 on the N-termini of A3 and B3, respectively. These amino acids are invisible in the cryo-EM density maps obtained in GDN and described above. The truncated A3 and B3 formed functional channels when co-transfected in HEK 293T cells and had an EC$_{50}$ of 20 µM for cGMP (Supplementary Fig. 6), which is similar to that of full-length WT A3/B3 (EC$_{50}$ = 22 µM) and the reported EC$_{50}$ of 12.9–15.8 µM of WT A3/B3 expressed in *Xenopus* oocytes[38,39]. This result indicates that the N-terminal truncations do not alter cGMP activation of A3/B3. As such, and for simplicity, we will hereafter continue to refer to the truncated CNGA3 and CNGB3 as A3 and B3. It has been shown that similar N-terminal deletions do not alter functional properties of CNGA1/CNGB1 channels[33]. Second, to facilitate the 3 A3:1 B3 stoichiometric assembly of the A3/B3 complex, A3 and B3 were cloned and expressed in a bicistronic construct (see "Methods" for details), following a strategy used for CNGA1/CNGB1[33]. Third, A3/B3 was reconstituted in nanodiscs with simply 1-palmitoyl-2-oleoyl-sn-glycero-3-phosphoglycerol (POPG) and 1-palmitoyl-2-oleoyl-sn-glycero-3-phosphocholine (POPC).

With these improvements and by using the same multistep cryo-EM analyses as in GDN, we obtained five single-particle reconstructions with 336,268 (36.6%), 304,034 (33.1%), 208,556 (22.7%), 15,177 (1.7%), and 39,888 (4.3%) particles, respectively (Supplementary Figs. 7–10). These reconstructions produced five distinct structures representing five different states. First, we successfully obtained a cGMP-bound open-state structure of A3/B3 (4.3% particles) at 3.33 Å resolution (Fig. 2a, Table 2, and Supplementary Figs. 7–10). This structure is named ND_cGMP_open. Second, a cGMP-bound closed-state structure, named ND_cGMP_closed (36.6% particles) was also obtained, at a resolution of 3.11 Å (Fig. 2b, Table 2, and Supplementary Figs. 7–10). This structure is

essentially identical to GDN_cGMP_closed, the closed-state structure of full-length A3/B3 (Fig. 2b), with an r.m.s.d. of 0.44 Å, indicating that the N-terminal truncations do not alter A3/B3 structure. Comparison of the structures of A3$_L$ in ND_cGMP_open and ND_cGMP_closed, with some key amino acids in the gating ring and S6 marked, shows large conformational rearrangements in the CNBD, the C-linker/gating ring, and S4-S6, including upward swinging of the C-helix of the CNBD, upward pushing of the gating ring, outward expansion of S4 and S5 away from the pore, and dilation and rotation of S6 (Fig. 2c). Some of these structural changes will be described in detail below. In general, these conformational changes are similar to those between the apo closed-state and cGMP-bound open-state structures of TAX-4, CNGA1, and CNGA1/CNGB1[28,29,33]. Third, two intermediate states and a pre-open state were captured, with resolution of 3.33 Å, 3.33 Å, and 3.60 Å, respectively (Fig. 2d, Table 2 and Supplementary Figs. 7–10). These states are named ND_cGMP_intermediate1, ND_cGMP_ intermediate2, and ND_cGMP_pre-open (with 33.1%, 22.7%, and 1.7% particles, respectively). Pore profile analysis taking into account of radius (Fig. 2e) and hydrophobicity[40] suggests that all three intermediate states are closed. However, the gate-forming residues (F392 and V396 in A3 and F434 and I438 in B3) adopt different positions and projections in some of these states, as described in detail later.

In brief, five distinct structures of cGMP-bound A3/B3 are obtained in POPG/POPC nanodiscs, representing closed, intermediate, pre-open, and open states. Strikingly, even though all four subunits of all the channels are bound with cGMP, most of the channels are in closed (36.6%) or intermediate states (55.8%), and only a very small fraction of the channels is in the pre-open (1.7%) or open states (4.3%). The distributions of the different states in GDN and POPG/POPC nanodiscs indicate that copurified native lipids, not GDN, arrest A3/B3 in the closed or intermediate states and POPG/POPC can liberate some channels from these states and enable them to open.

## Conformational change wave in the CNBD and C-linker/gating ring

Ensemble and pairwise comparisons of the five structures obtained in POPG/POPC nanodiscs show that A3/B3 undergoes subunit-unique, graded, and localized conformational rearrangements during the sojourn from closed to open (Figs. 3 and 4 and Supplementary Movie 2). These conformational rearrangements originate from the CNBD, propagate through the C-linker/gating ring to the TMD, and eventually lead to dilation and rotation of S6 and opening of the cavity gate and arginine gate. To illustrate the incremental conformational rearrangements in the CNBD and C-linker/gating ring, we select the A3$_D$/A3$_R$ "elbow-on-shoulder" pair since they exhibit the most prominent structural changes between the closed and open states and in the intermediate states (Fig. 3a, b). To facilitate visualization of the structural changes, the Cα positions and distance changes of some selected amino acids in several key or representative regions are marked (Fig. 3a, b). In the closed state, the C-helix of the CNBD is in a "down" position. In the open state, it moves to an "up" position, rotates clockwise with respect to the TMD by ~15° (viewed from the outside) (Fig. 2c), and rotates along its axis by ~105° (Figs. 1c and 3a). Many other parts also move toward the TMD, including helices AB of the CNBD, helices E'F' of the C-linker, and helices A'B'C'D' of the gating ring. Helix A' also undergoes other complex motions, such as tilting away from the center and rotating along its axis. These movements increase the distance between two diagonally opposed N407 (in A3) or T449 (in B3), which is connected to S6 (Fig. 2c), from 22.5 Å to 29.0 Å (Cα to Cα) between A3$_D$ and B3, and from 22.2 Å to 26.7 Å between A3$_L$ and A3$_R$. This expansion naturally leads to dilation of S6.

The conformational rearrangements in the CNBD and C-linker/gating ring of the intermediate and pre-open states are incremental and much subtler, as illustrated by pairwise comparisons (Fig. 3b). Take helix C and helix A' as examples. In ND_cGMP_intermediate1, both

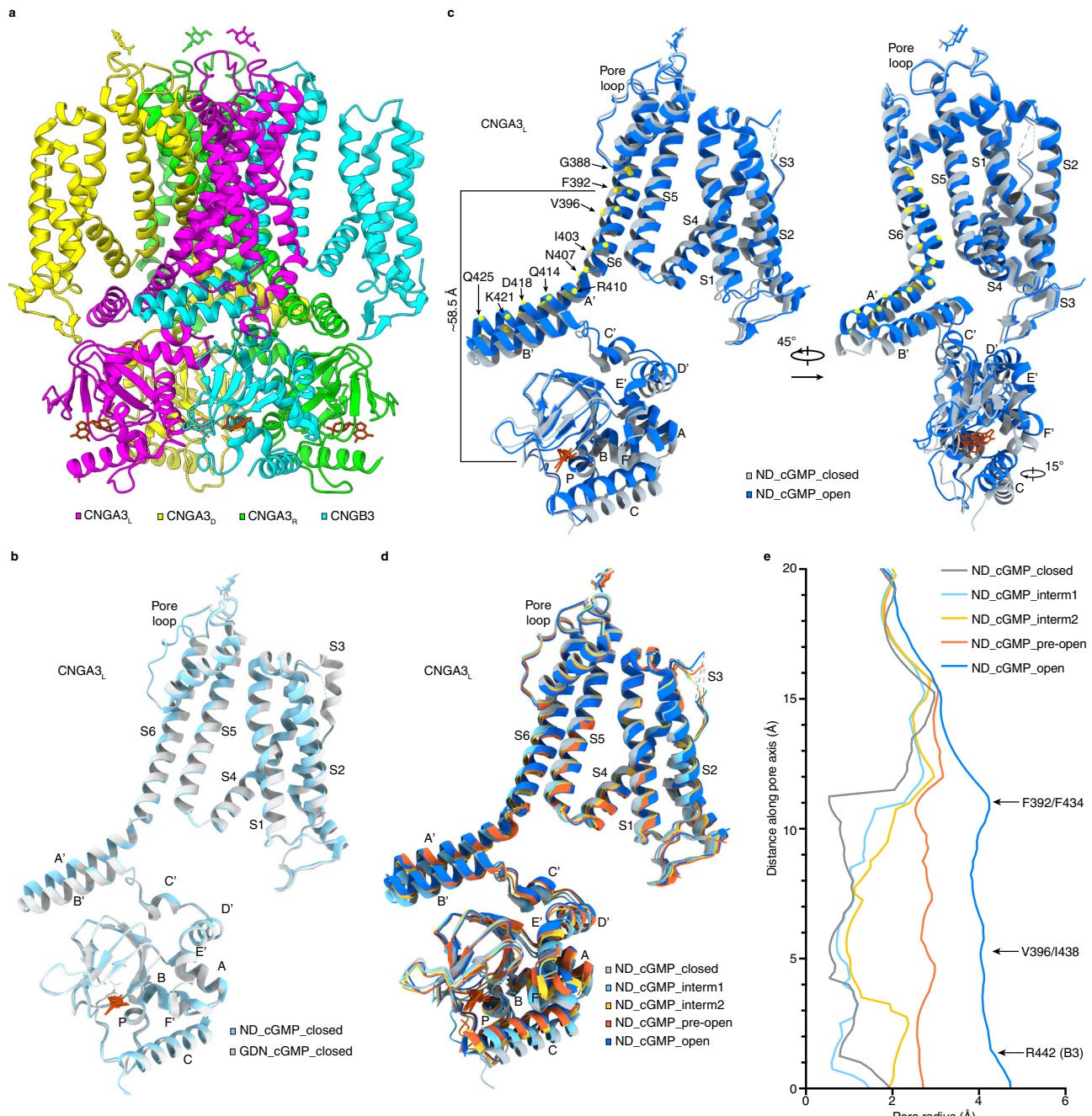

**Fig. 2 | Cryo-EM structures of cGMP-bound CNGA3/CNGB3 in lipid nanodisc.**
**a** Open-state structure, viewed parallel to the membrane. **b** Superposition of CNGA3$_L$ protomer of GDN_cGMP_closed and ND_cGMP_closed. **c** Superposition of CNGA3$_L$ protomer of ND_cGMP_closed and ND_cGMP_open. The Cα positions of selected key residues on S6 and A' are shown as yellow dots. **d** Superposition of CNGA3$_L$ protomer of all five states of liganded CNGA3/CNGB3 in lipid nanodisc. **e** Pore-size profiles of the solvent-accessible pathway of the five liganded states of CNGA3/CNGB3 in lipid nanodisc. The origin of the pore axis is set at the intracellular end of S6, just below R442 of CNGB3.

helices move upward but the C-terminal ends move more than the N-terminal ends do. In ND_cGMP_intermediate2, helix C moves up further (particularly its N-terminal end), rotates along its own axis, and slides leftward; the C-terminal end of helix A' also moves up further. In ND_cGMP_pre-open, helix C undergoes further rotation and helix A' moves further upward and tilts away from the center. These movements are in the expected direction toward channel opening.

### Incremental S6 movement and pore opening

The cavity gate, formed by F392 and V396 in A3 and F434 and I438 in B3, is located ~58.5 Å from the cGMP binding site (Fig. 2c).

The conformational changes in the CNBD and C-linker/gating ring must be transmitted to the TMD to open this gate. There are two coupling mechanisms in this allosteric process. First, helix A' of the gating ring pulls S6, which is connected to the helix A' via a two-amino acid linker (M406 and N407) (Fig. 2c). Thus, helix A' movements directly cause S6 to move (Fig. 3d). Second, helices A'B' of the gating ring of one subunit interacts with S4-S5 of its clockwise adjacent subunit via hydrogen bonds and salt bridges (Fig. 4). These interactions vary between different subunit pairs and in different states (Fig. 4f). In the closed state and intermediate state 1, R410 of A3 and R456 of B3 are the only amino acids on the gating ring (i.e., helices A'B') that interact

**Table 2 | Cryo-EM data collection, refinement, and validation of nanodisc samples**

| | ND_closed (EMDB-28622) (PDB 8EV8) | ND_interm1 (EMDB-28623) (PDB 8EV9) | ND_interm2 (EMDB-82624) (PDB 8EVA) | ND_pre-open (EMDB-82625) (PDB 8EVB) | ND_open (EMDB-82626) (PDB 8EVC) |
|---|---|---|---|---|---|
| **Data collection and processing** | | | | | |
| Magnification | 105,000 | 105,000 | 105,000 | 105,000 | 105,000 |
| Voltage (kV) | 300 | 300 | 300 | 300 | 300 |
| Electron exposure (e$^-$/Å$^2$) | 61.05 | 61.05 | 61.05 | 61.05 | 61.05 |
| Defocus range (µm) | −0.9 to −1.5 | −0.9 to −1.5 | −0.9 to −1.5 | −0.9 to −1.5 | −0.9 to −1.5 |
| Pixel size (Å) | 0.83 | 0.83 | 0.83 | 0.83 | 0.83 |
| Symmetry imposed | C1 | C1 | C1 | C1 | C1 |
| Initial particle images (no.) | 5,265,196 | 5,265,196 | 5,265,196 | 5,265,196 | 5,265,196 |
| Final particle images (no.) | 336,268 | 304,034 | 208,556 | 15,177 | 39,888 |
| Map resolution (Å) | | | | | |
| FSC threshold | 0.143 | 0.143 | 0.143 | 0.143 | 0.143 |
| Map resolution range (Å) | 2.63 to 5.97 | 2.85 to 6.65 | 2.80 to 6.53 | 3.29 to 39.13 | 2.95 to 8.03 |
| **Refinement** | | | | | |
| Initial model used (PDB code) | 7RHS | 7RHS | 7RHS | 7RHS | 7RHS |
| Model resolution (Å) | 3.13 | 3.36 | 3.36 | 3.70 | 3.40 |
| FSC threshold | 0.143 | 0.143 | 0.143 | 0.143 | 0.143 |
| Model resolution range (Å) | n/a | n/a | n/a | n/a | n/a |
| Map sharpening $B$ factor (Å$^2$) | −117.1 | −130.1 | −123.5 | −60.1 | −78.8 |
| Model composition | | | | | |
| Non-hydrogen atoms | 14,748 | 14,661 | 14,591 | 14,456 | 14,531 |
| Protein residues | 1789 | 1778 | 1768 | 1751 | 1764 |
| Ligands | 4 | 4 | 4 | 4 | 4 |
| $B$ factors (Å$^2$) | | | | | |
| Protein | 27.93 | 68.31 | 50.93 | 98.40 | 68.00 |
| Ligand | 47.68 | 52.75 | 54.61 | 64.17 | 53.60 |
| R.m.s. deviations | | | | | |
| Bond lengths (Å) | 0.004 | 0.010 | 0.006 | 0.008 | 0.005 |
| Bond angles (°) | 0.639 | 1.284 | 0.834 | 1.361 | 0.892 |
| Validation | | | | | |
| MolProbity score | 1.50 | 1.71 | 1.65 | 1.83 | 1.63 |
| Clashscore | 6.25 | 8.02 | 8.06 | 9.82 | 6.48 |
| Poor rotamers (%) | 0.93 | 0.38 | 0.82 | 0.63 | 0.95 |
| Ramachandran plot | | | | | |
| Favored (%) | 97.13 | 96.03 | 96.69 | 95.49 | 96.05 |
| Allowed (%) | 2.87 | 3.97 | 3.31 | 4.51 | 3.95 |
| Disallowed (%) | 0 | 0 | 0 | 0 | 0 |

with the TMD, mainly D289 and E292 on S4. As the channel transits from the closed state to the open state, the number and hence strength of gating ring-TMD interactions increase gradually (Fig. 4f) due to the progressive pushup, tilting, and rotation of helices A′B′ (Fig. 4a–e). In the pre-open and open states T295 and N296 on the S4-S5 linker and R302 on S5 and many residues on helices A′B′ are recruited, with more and stronger interactions in the open state than in the pre-open state (Fig. 4d–f). As a result of such graded interactions, S4 and S5 gradually move away from the center (Fig. 3c), which would cause S6 to dilate owing to the intimate interactions between S5 and S6. This structural implication of S5 in CNG channel gating is consistent with previous work showing that mutations in and/or modification of S5 greatly affect activation of CNGA1[41].

As a result of direct pulling by helix A′ and outward expansion of S5 and S4, S6 undergoes incremental dilation and rotation, especially its C-terminal half (Fig. 3d), which lead to a gradual opening of the cavity gate (Fig. 5). As in GDN, the gate residue F392 of A3$_L$ has a sideway projection and a pore projection with similar occupancy in

ND_cGMP_closed, ND_cGMP_intermediate1, ND_cGMP_intermediate2 and ND_cGMP_pre-open, whereas F392 of A3$_D$ and A3$_R$, F434 of B3, V396 of A3 and I438 of B3 have only pore projection in all these states (Supplementary Figs. 3d–g and 4d–g). Compared to ND_cGMP_closed, there is little or no movement of the gate residues (F392 and V396 of A3 and F434 and I438 of B3) in ND_cGMP_intermediate1 and ND_cGMP_intermediate2, except a modest reorientation of the A3$_D$ F392 side-chain; the cavity gate is thus tightly closed in both intermediate states (Fig. 5a, b, e). In ND_cGMP_pre-open, F392 and V396 in A3$_R$ remain unchanged, but the other gate residues move away from the center of the pore by varying distances and angles (Fig. 5c, f). The side-chains of V396 in A3$_L$ and A3$_D$ and I438 in B3 rotate away from the pore, significantly widening this part of the pore and making it largely open (Fig. 5f). However, the side-chains of F392 in A3$_D$ and A3$_R$ and F434 in B3 still project to the pore (Fig. 5c) and still form a hydrophobic gate[40]. Thus, ND_cGMP_pre-open is still closed. In ND_cGMP_open, the side-chains of all gate residues, including F392 in A3$_L$, project away from the pore, rendering the cavity gate fully open (Fig. 5d, g and Supplementary

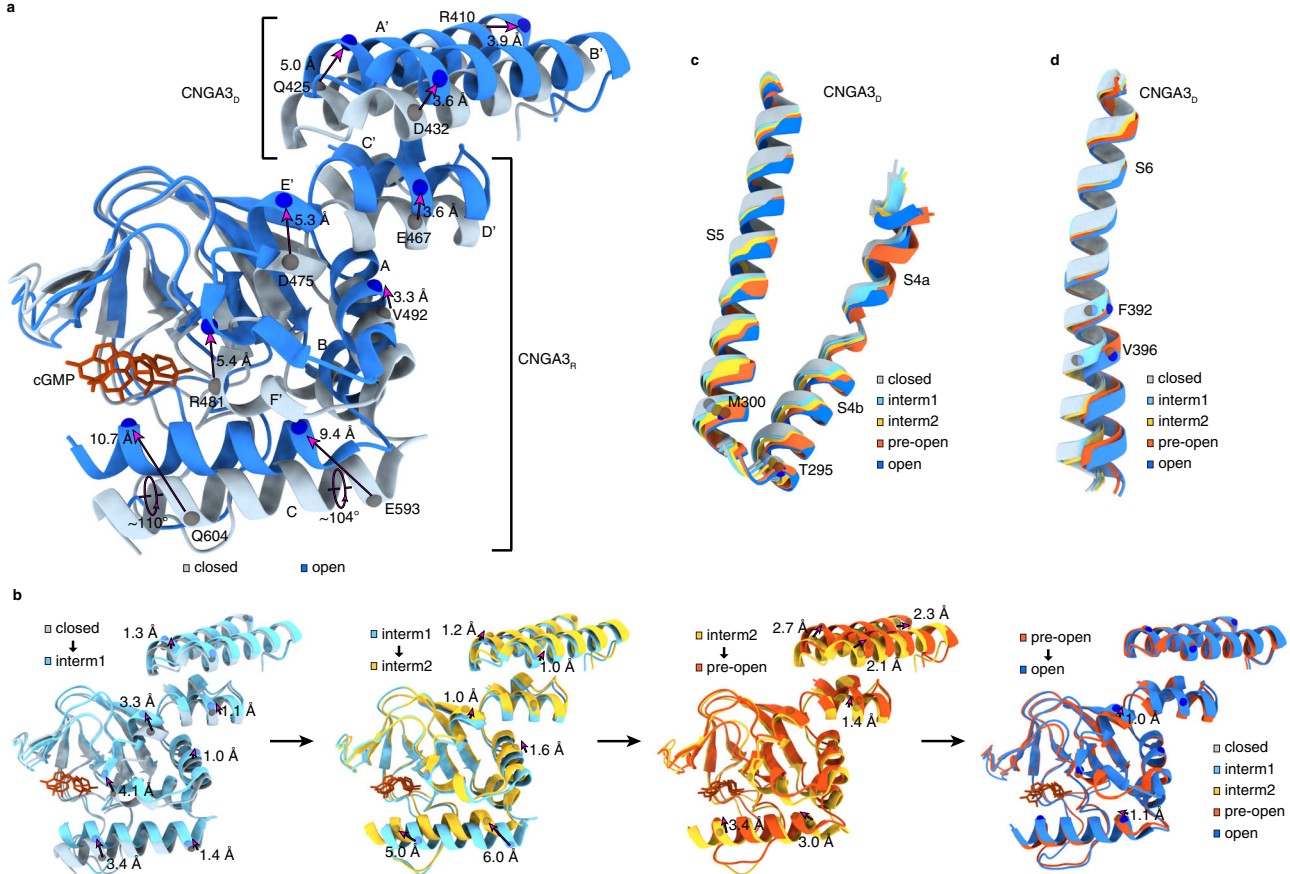

**Fig. 3 | Incremental conformational changes in the CNBD, C-linker/gating ring and S4-S6. a** Superposition of the CNBD and C-linker/gating ring regions of CNGA3$_R$ and CNGA3$_D$ in closed and open states. The Cα movement of selected residues are indicated by magenta arrows. **b** Pairwise superposition of the CNBD and C-linker/gating ring regions of CNGA3$_R$ and CNGA3$_D$ in closed versus intermediate 1, intermediate 1 versus intermediate 2, intermediate 2 versus pre-open, and pre-open versus open states. The Cα movement of the same residues as in (**a**) are indicated with magenta arrows. **c** Superposition of S4-linker-S5 of CNGA3$_D$ in all five states. The Cα positions of M300 in S5 and T295 in S4 are marked with dots. **d** Superposition of S6 of CNGA3$_D$ in all five states. The Cα positions of gate-forming F392 and V396 are marked with dots.

Figs. 3h and 4h). It is notable that the conformation of A3$_R$ S6 remains largely unchanged throughout the gating transitions (Fig. 5a–c, e, f) and its gate residues open up only in the open state (Fig. 5d and g), reinforcing the notion of subunit-unique gating transitions.

The arginine gate formed by R442 of B3 also undergoes gradual conformational changes during closed-to-open transitions and opens fully only in the open state (Fig. 6). In ND_cGMP_closed and ND_cGMP_intermediate1, R442 projects to the ion conduction pathway with up and down orientations (Fig. 6a, b), as in GDN. The up orientation diminishes in ND_cGMP_intermediate2 and ND_cGMP_pre-open (Fig. 6c, d). In ND_cGMP_open, the R442 side-chain swings from the center of the pore to the side and opens the arginine gate (Fig. 6e, f). R442 is engaged in different interactions with S6 of A3$_L$ and A3$_R$ in different states (Fig. 6a–e). In ND_cGMP_closed, R442 interacts with V396 of A3$_R$ in the up orientation but with S404 of A3$_L$ in the down orientation (Fig. 6a). In ND_cGMP_intermediate1, the R442-V396 interaction is maintained but the R442-S404 interaction is lost (Fig. 6b). Both interactions are lost in ND_cGMP_intermediate2, ND_cGMP_pre-open and ND_cGMP_open (Fig. 6c–e), but in the open state R442 is engaged in a new interaction as a result of its rotation, forming a strong hydrogen bond with G397 of A3$_L$ (Fig. 6e). Rotamer conformations and interactions with S6 of neighboring subunits have also been observed for the arginine gate in rod CNGA1/CNGB1 channels[30,31,33,34]. The functional importance, if any, of a flexible arginine gate in cone and rod photoreceptor CNG channels remains to be elucidated[31].

## Lipid binding

The density maps of cGMP-bound A3/B3 in GDN and POPG/POPC nanodiscs are heavily decorated with nonprotein densities (Fig. 7a–d). Some of these densities are likely noise but some probably represent tightly associated, copurified endogenous lipids. Indeed, mass spectroscopy of a purified A3/B3 protein sample reconstituted in GDN shows the presence of many different types native lipids, including cholesterol, phosphatidylserine, phosphatidylethanolamine, phosphatidylcholine, ceramide and diacylglycerol (DAG). It is difficult to model specific lipid molecules into nonprotein densities in most cases due to limited local resolution and/or heterogeneity of lipid occupation, but in some cases candidate lipids appear to fit the densities reasonably well (Fig. 7a–d). Definitive lipid modeling, however, would require better local resolution and confirmation by mutagenesis experiments.

## Discussion

We obtained three distinct structures of cGMP-bound A3/B3 in GDN and five distinct structures of cGMP-bound A3/B3 in POPG/POPC nanodiscs; of these, five structures capture distinct intermediate gating states (two in GDN and three in POPG/POPC nanodiscs). These structures represent a spectrum of snapshots of A3/B3 in different states, from closed to intermediate to pre-open to open, providing a comprehensive view of the conformational trajectory of allosteric gating of a vertebrate CNG channel by its natural ligand. This process is schematized in Fig. 8, which shows in greater detail than before[26,28,29,33]

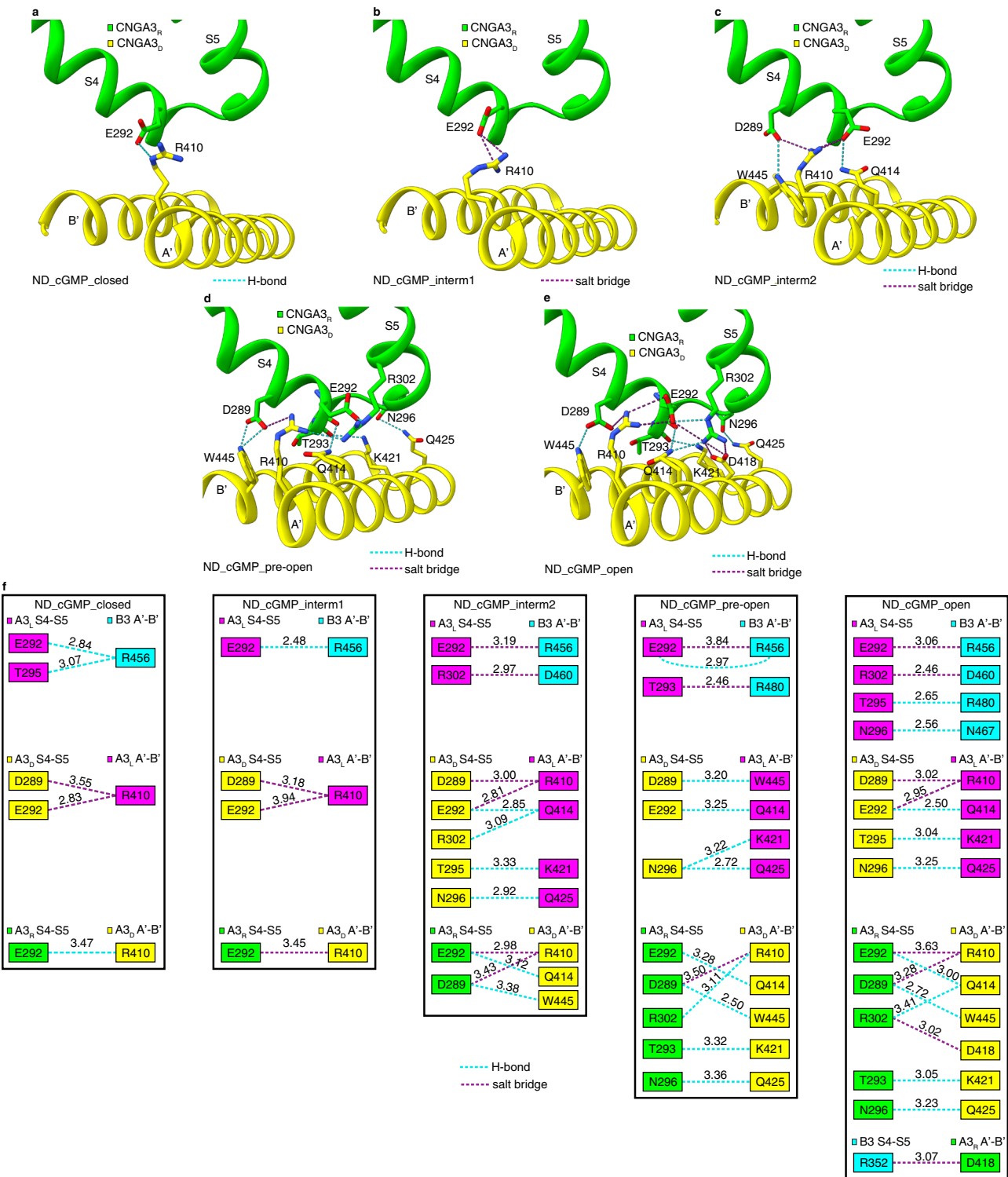

**Fig. 4 | State- and subunit-dependent interactions between the gating ring and TMD. a–e** Close-up view of local structures and amino acids of CNGA3 A'-B' and CNGA3$_R$ S4-S5 in closed state (**a**), intermediate state 1 (**b**), intermediate state 2 (**c**), pre-open state (**d**), and open state (**e**). For clarity, only amino acids participating in intersubunit hydrogen bonding and salt bridges are shown as sticks. **f** The five different states of cGMP-bound CNGA3/CNGB3 obtained in POPG/POPC nanodiscs are boxed. In each box, residues of S4-S5 are listed on the left and residues of helices A'B' of the gating ring are listed on the right. Each residue is color coded based on its subunit origin, using the color scheme of Fig. 2a. Intersubunit hydrogen bonds and salt bridges are indicated with cyan and purple dashed lines, respectively. Numbers indicate interaction distances in Angstrom.

a presumed sequence of some of the key conformational arrangements triggered by the binding of cGMP to the CNBD of the apo channel: (1) The C-terminal end of C-helix of the CNBD moves toward cGMP; (2) The N-terminal end of C-helix moves toward the membrane, and the entire C-helix slides radially and rotate with respect to the TMD; (3) Helices E'F' of the C-linker move toward the membrane, causing helices A'B'C'D' of the gating ring to move upward and form stronger interactions with S4 and S5; (4) S4, S5 and helix A' of the

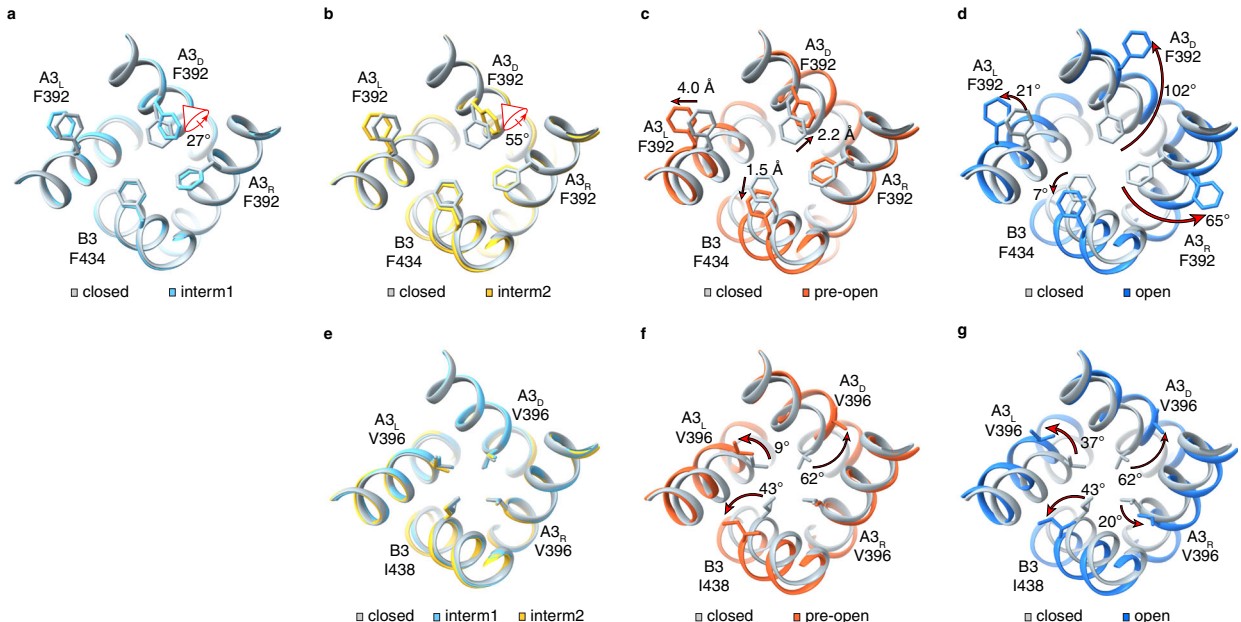

**Fig. 5 | Gradual opening of the cavity gate. a–d** Pairwise comparison of S6 in closed versus intermediate 1 (**a**), closed versus intermediate 2 (**b**), closed versus pre-open (**d**), and closed versus open (**e**) states, viewed from extracellular side. Motions of the side-chains of gate-forming F392/F434 are indicated. F392 of A3$_L$ has a pore projection and a sideway projection in closed and intermediate 1 states but only the sideway projection is shown. **e–g** Comparison of S6 in closed versus intermediate 1 and intermediate 2 (**e**), closed versus pre-open (**f**), and closed versus open (**g**) states, viewed from extracellular side. Motions of the side-chains of gate-forming V396/I438 are indicated.

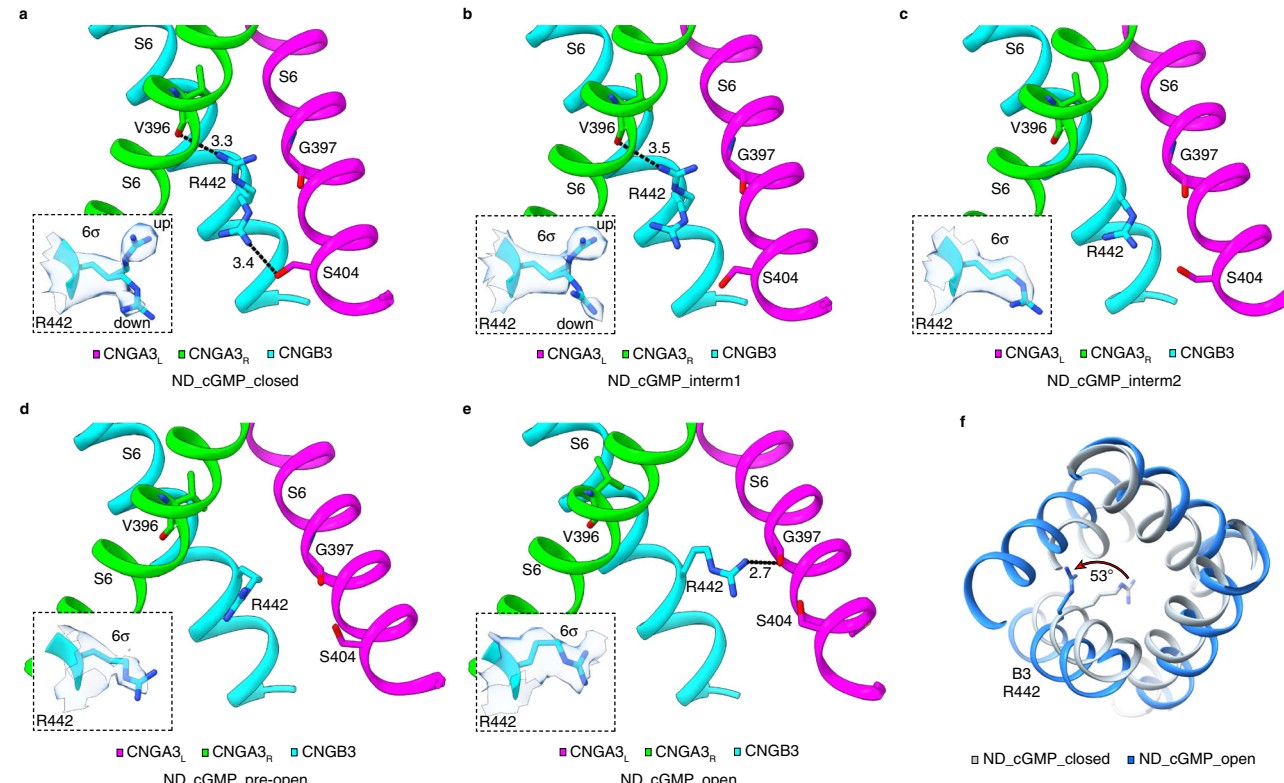

**Fig. 6 | Projections, interactions, and opening of the arginine gate. a–e** State-dependent projections and interactions of R442 of CNGB3 in closed (**a**), intermediate 1 (**b**), intermediate 2 (**c**), pre-open (**d**), and open (**e**) states. R442 side-chain exhibits up and down rotamer conformations in closed (**a**) and intermediate 1 (**b**) states, and projects directly to the pore in (**a–d**) and to the side in (**e**). Dashed lines represent hydrogen bonds, with indicated distances (in Angstrom). Insets show densities of R442 (all contoured to 6σ) as transparent blobs and superimposed with the modeled amino acid. **f** Superposition of S6 (viewed from extracellular side) in closed and open states, highlighting the movement of R442 of CNGB3.

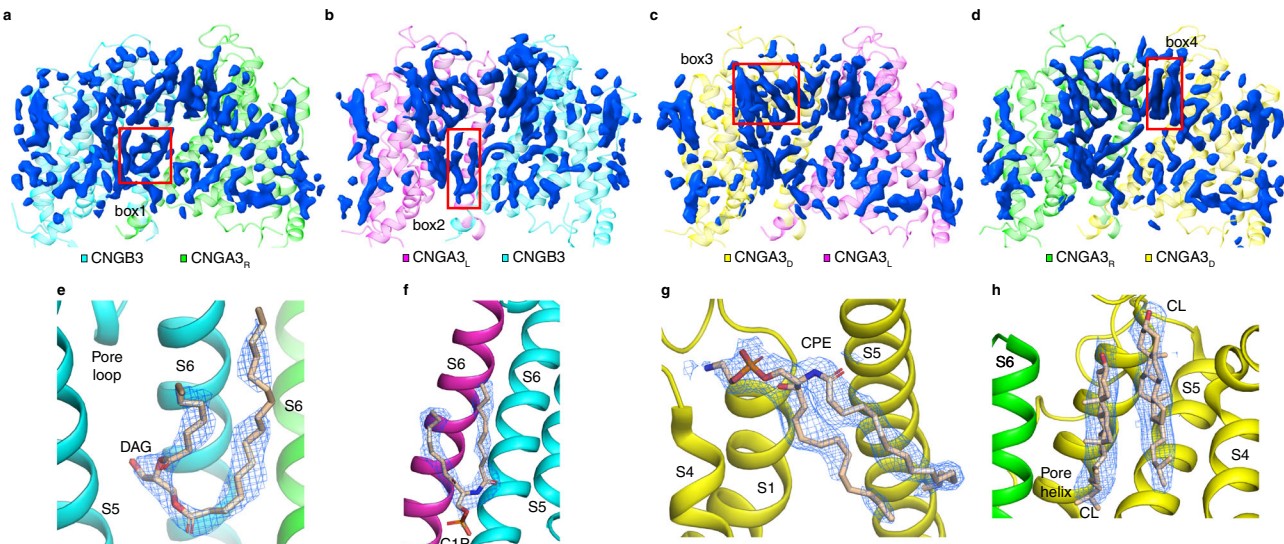

**Fig. 7 | Putative binding of native lipids to CNGA3/CNGB3. a–d** Profiles of non-protein densities (blue) in the indicated CNGA3/CNGB3 subunits and intersubunit interfaces in the absence of cGMP and in GDN. **e–h** Close-up views of putative lipid binding in the boxed regions in (**a–d**). Candidate lipids are built based on the location and shape of the densities. Densities are shown as blue meshes and are contoured at 1.5σ in (**e**), 3.5σ in (**f**), 2.5σ in (**g**), and 2σ in (**h**). DAG: diacylglycerol; C1P: ceramide-1-phosphate; CPE: ceramide phosphoethanolamine; CL: cholesterol. Source data of MS-based lipidomics and all identified lipid species are available in the Source data file.

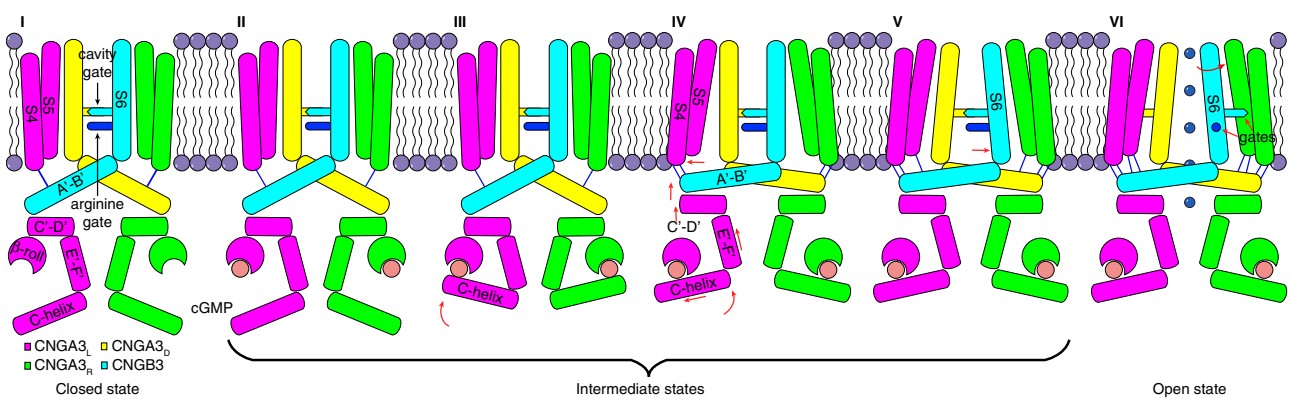

**Fig. 8 | Model of allosteric gating of CNGA3/CNGB3 by cGMP.** The schematic, based on the closed, intermediate, pre-open and open state structures obtained in POPG/POPC nanodiscs, depicts stepwise conformational rearrangements in several key regions during channel opening. The conformational changes triggered by cGMP binding to the β-roll and C-helix of the CNBD propagate to the pore-forming S6 through helices E'F' of the C-linker and helices A'B'C'D' of the gating ring, ultimately resulting in the opening of the cavity gate and arginine gate. The sequential movement of different regions is highlighted in red. Blue lines indicate interactions between S4-S5 and helices A'B'.

gating ring move away from the pore axis, causing the C-terminal end of S6 to expand radially; (5) S6 dilates further and rotates along its own axis, causing both the cavity gate and arginine gate to fully open. The actual subunit-specific conformational rearrangements are of course much more complex and can be more clearly visualized and conceptualized in a movie generated from the 3D variability analysis (Supplementary Movie 2), which shows a continuum of 13 different 3D maps generated from the vast majority of the particles used to produce the five states in POPG/POPC nanodiscs.

There are notable similarities and differences between the structures of A3/B3 in different states and those of CNGA1/CNGB1[33] and SthK Y26F and R120A mutants[7]. In the fully open-state of A3/B3, S6 and the cavity gate-forming residues of all four subunits undergo large conformational changes (Fig. 5d, g). However, in the two open-state structures of CNGA1/CNGB1, S6 and the cavity gate-forming residues of two subunits (CNGA1$_R$ and CNGB1) show little or only very small conformational changes[33], suggesting that these structures actually represent partially open channels[42]. It is postulated that co-purified calmodulin plays a role in stabilizing CNGA1/CNGB1 in these partially open states[42]. This hypothesis can now be tested since a structure of CNGA1/CNGB1 in complex with Ca$^{2+}$/calmodulin has recently been solved[43]. Calmodulin was unlikely bound to our A3/B3 sample because 2 mM EGTA was included during protein purification, and no density corresponding to calmodulin was observed in any density maps. Two partially open states are observed in SthK R120A, both of which have an activated CNBD and C-linker/gating ring and an open but narrower pore[7]. We did not obtain a partially open-state structure of A3/B3, presumably because such a state is energetically unstable under our experimental conditions. On the other hand, the intermediate states described here for A3/B3 are not observed in CNGA1/CNGB1 and SthK mutants. A pre-open state is observed in SthK Y26F[7]. It is similar to the pre-open state of A3/B3 in that both have a nearly fully activated CNBD and C-linker/gating ring but a closed pore, yet they are different in that there are large conformational rearrangements in S6 and the cavity

gate-forming residues in A3/B3 (Fig. 5c, f) but not in SthK Y26F. Lastly, the closed-open conformational rearrangements of A3/B3 vary from subunit to subunit, as has been observed for CNGA1/CNGB1[33]. This variation is most vividly demonstrated by the gradual and differential movement of the cavity gate-forming residues (Fig. 5). The root cause of this asymmetry is likely the presence of B3, which results in intrinsically different interactions among the four subunits, as exemplified by the state-dependent interactions between S4-S5 and helices A′B′ of the gating ring (Fig. 4).

The activities of many ion channels are modulated by lipids, and there is growing exploration and understanding of the structural basis of such modulation[44,45]. Abundant nonprotein densities presumably representing lipids are observed in cryo-EM density maps of TAX-4, CNGA1, and CNGA1/CNGB1[28–30,33]. Lipids are also present in SthK structures and modulate SthK activity[27,32]. In our study, three observations in particular suggest that lipids play a key role in shaping the energetic landscape of cGMP activation of A3/B3. (1) All A3/B3 channels in GDN and most (>95%) channels in POPG/POPC nanodiscs are in the closed, intermediate, and pre-open states, even though the channels are fully bound with cGMP. (2) The open state exists only in POPG/POPC nanodiscs. (3) Numerous nonprotein densities are observed and a variety of lipids, including DAG, are detected in cGMP-bound A3/B3 protein samples reconstituted in GDN nanodiscs. Previous studies have shown that DAG, phosphatidylinositol (4,5)-trisphosphate, and phosphatidylinositol (3,4,5)-trisphosphate inhibit native and heterologously expressed CNG channels, probably through direct binding to the channels[39,46–51]. We speculate that A3/B3 is inhibited by copurified DAG and that POPG/POPC can displace some bound DAG. Building on the structures presented in this work, it will be interesting to structurally, functionally, and dynamically study the effects of DAG and other lipids on CNG channel gating, including using tools such as single molecule fluorescence resonance energy transfer[7] and time-resolved transition metal ion fluorescence resonance energy transfer[52–54]. It will also be interesting to examine whether and how disease-associated mutations in photoreceptor CNG channels affect lipid interactions with the channels.

## Methods

### Molecular cloning

cDNAs encoding human CNGA3 (NCBI Reference Sequence: NP_001289.1) and CNGB3 (NCBI Reference Sequence: NP_061971.3) were amplified by RT-PCR from HEK 293T cells (ATCC) using SuperScript™ III First-Strand Synthesis System (ThermoFisher 18080051). For protein expression and purification, full-length CNGA3 was cloned into the pEZT-BM vector (Addgene #74099)[55] with an N-terminal MBP tag followed by a P3C cleavage site and a C-terminal FLAG tag; full-length CNGB3 was cloned into the same vector with an N-terminal 2× Strep tag. To increase protein expression, full-length CNGA3 and CNGB3 were both N-terminally truncated for reconstitution into POPG/POPC nanodiscs. The truncated CNGA3 (amino acids 152-694) and CNGB3 (amino acids 79–809) were both tagged with N-terminal FLAG and cloned into the same pEZT-BM vector connected by a "self-cleaving" P2A peptide linker.

### Protein expression, purification, and nanodisc reconstitution

CNGA3/CNGB3 were expressed in HEK 293S GnTi⁻ cells (ATCC) using the BacMam system. Recombinant baculoviruses were generated with Sf9 insect cells (Expression Systems) following the standard protocol (full-length CNGA3 and CNGB3 virus were generated separately). Harvested baculoviruses were amplified twice in Sf9 cells to obtain sufficient viruses for large-scale infection. HEK 293S GnTi⁻ cells were grown in suspension at 37 °C. When cells reached a density of $2$–$2.5 \times 10^6$ cells mL⁻¹, P2 baculovirus was added to the culture (10%, v/v). For full-length CNGA3 and CNGB3, a volume ratio of 1:1.5 CNGA3:CNGB3 P2 virus was used for infection. After incubation

for 12–24 h, the culture was supplemented with 10 mM sodium butyrate to boost the expression, and was further incubated at 30 °C for 72 h before collection.

Purification of CNGA3/CNGB3 was carried out at 4 °C. Cell pellet from 6.4 L culture was resuspended and lysed by stirring for 30 min in 200 mL hypotonic buffer (10 mM HEPES-Na, pH 8.58, 0.1 mM TCEP, 2 mM EGTA) supplemented with a protease inhibitor cocktail (Millipore Sigma S8830). Membrane fraction was collected by centrifugation at $29,448 \times g$ for 40 min, and then homogenized with a Dounce homogenizer in 200 mL extraction buffer (50 mM HEPES-Na, pH 8.58, 150 mM NaCl, 2% DDM, 0.2% CHS, 0.1 mM TCEP, 2 mM EGTA). After incubation for 2 h, the solubilized membrane was clarified by ultracentrifugation at $111,338\,g$ for 50 min.

For full-length CNGA3/CNGB3, the supernatant was incubated with amylose resin (New England Biolabs) for 2 h with gentle agitation. The resin was collected by low-speed spin at $800 \times g$, transferred into a gravity column, and washed with 20 CV wash buffer A (50 mM HEPES-Na, pH 8.58, 150 mM NaCl, 0.1% DDM, 0.01% CHS, 0.1 mM TCEP, 2 mM EGTA), followed by 30 CV wash buffer B (50 mM HEPES-Na, pH 8.58, 150 mM NaCl, 0.02% GDN, 0.1 mM TCEP, 2 mM EGTA). The protein was eluted from amylose resin with 30 mL wash buffer B containing 20 mM maltose, and then loaded onto a 1 mL StrepTrap HP column (Cytiva 28-9075-46). The column was washed with 20 CV of wash buffer B before eluted with 20 mL wash buffer B containing 5 mM d-Desthiobiotin. Eluted protein was incubated with P3C protease overnight to cleave the MBP tag and was further purified by a Superose 6 increase column (Cytiva 29-0915-96) equilibrated with a gel-filtration buffer (50 mM HEPES-Na, pH 8.58, 150 mM NaCl, 0.02% GDN, 0.1 mM TCEP, 2 mM EGTA). Fractions corresponding to hetero-tetrameric CNGA3/CNGB3 were pooled and concentrated to 1.5 mg mL⁻¹. cGMP was added before grid freezing to a final concentration of 2 mM.

For truncated CNGA3/CNGB3, the supernatant was incubated with Anti-FLAG M2 resin (Sigma A2220) for 2 h with gentle agitation. The resin was collected by low-speed spin at $800 \times g$, transferred into a gravity column, and washed with 20 CV wash buffer (HEPES-Na, pH 8.58, 150 mM NaCl, 0.1% DDM, 0.01% CHS, 0.1 mM TCEP, 2 mM EGTA). The protein was eluted from Anti-FLAG resin with 5 mL wash buffer containing 200 μg mL⁻¹ FLAG peptide (Bio-synthesis). The eluted CNGA3/CNGB3 complex was concentrated to 1 mL before nanodisc reconstitution.

Membrane scaffold protein MSP2N2 was expressed and purified from *Escherichia coli*. CNGA3/CNGB3 complex, MSP2N2, and lipids (1:1 POPC and POPG) were mixed at a molar ratio of 1:5:250 and incubated on ice for 1 h. 400 mg Bio-beads SM2 (Bio-Rad 1528920) were added to initiate the reconstitution by removing detergents from the system, and the mixture was incubated at 4 °C with gentle agitation overnight. The next morning, the mixture was centrifuged to remove Bio-beads and concentrated to 0.5 mL before the final purification by a Superose 6 increase column (Cytiva 29-0915-96) equilibrated with a gel-filtration buffer (50 mM HEPES-Na, pH 8.58, 150 mM NaCl, 0.1 mM TCEP, 2 mM EGTA). Fractions corresponding to hetero-tetrameric CNGA3/CNGB3 were pooled and cGMP was added to the sample to a concentration of 2 mM. The pooled fractions were concentrated to 2 mg mL⁻¹ and cGMP was added again before grid freezing to a final concentration of 4 mM.

### Cryo-EM sample preparation and data acquisition

Cryo-EM grids were prepared by applying 3 μL of CNGA3/CNGB3 sample to a glow-discharged UltrAuFoil R0.6/1 300 mesh gold grid (Quantifoil Micro Tools Q34953). After waiting for 10 s, the grid was blotted for 10 s (double-sided, blot force 3) under 100% humidity and 4 °C using FEI Vitrobot Mark IV (FEI) and immediately plunged into liquid ethane cooled by liquid nitrogen. Micrographs were acquired by a Titan Krios microscope (ThermoFisher) operated at 300 kV and equipped with a K3 direct electron detector (Gatan) working at counting mode. Leginon[56] was used for data collection. For the sample

of full-length CNGA3/CNGB3 in GDN, a nominal magnification of ×81,000 was used, corresponding to a final pixel size of 1.0582 Å on image. The defocus ranged from −0.9 μm to −1.9 μm. A total of 10,258 movies were collected. Each micrograph was dose-fractionated to 44 movie frames under a dose rate of 22.33 counts per pixel per second, with a total exposure time of 2.2 s and a frame exposure time of 50 ms, resulting in a total dose of 55.03 e⁻/Å². 

For the sample of truncated CNGA3/CNGB3 in POPG/POPC nanodiscs, a nominal magnification of ×105,000 was used, corresponding to a final pixel size of 0.8290 Å on image. The defocus ranged from −0.9 μm to −1.5 μm. A total of 7689 movies were collected. Each micrograph was dose-fractionated to 40 movie frames under a dose rate of 21.0 counts per pixel per second, with a total exposure time of 2 s and a frame exposure time of 50 ms, resulting in a total dose of 61.05 e⁻/Å².

### Image processing

For the dataset of full-length CNGA3/CNGB3 in GDN, drift correction, beam-induced motion correction, and dose-weighting were performed on 10,258 movies with MotionCor2[57] implemented in RELION 3.0[58]. Contrast-transfer function parameters of the micrographs were estimated using Gctf[59]. Particles were automatically picked by RELION 3.0. C1 symmetry was applied during all steps of data processing. Initially, 11,977,477 particles were picked by auto-picking, 4 × binned (4.2328 Å), and then classified by three rounds of 2D classification. After 2D classification, 1,770,574 particles corresponding to intact particles with evident secondary structures were selected. The selected particles were imported into cryoSPARC v3.3.2[37] for 2 rounds of Ab-initio reconstruction and heterogeneous refinement (6 classes). One class with 364,083 particles was selected, re-extracted, 2 × binned (2.1164 Å), and then applied to another round of heterogeneous refinement using volumes generated by previous ab-initio jobs (5 classes). Two classes of a total of 260,699 particles were selected and re-extracted without binning (1.0582 Å) and used for consensus refinement (non-uniform refinement). The resulting 3D map with a resolution of 3.60 Å was used to generate two separate masks. First, a mask was put around B3 only and was used as the focus mask for one round of 3D classification to determine if there was misalignment of the particles for the consensus map. The number of classes was set to 10, the target resolution was set to 4, and the online EM batch size was set to 5000. The resulting 10 classes were indeed all B3, judging by the presence of the N339 glycosylation site, the overall shape of the TMD, and the length and angle of the C helix. This analysis confirmed that no A3 was aligned with B3 and that all particles were properly aligned. Second, a mask enclosing the whole CNGA3/CNGB3 channel without detergent micelle density was generated using UCSF Chimera[60]. 3D-variability analysis was carried out using this mask, with a user setting of three modes and a filter resolution of 4 Å. 3D-variability display (simple) was used to examine each mode. Two of the three modes were similar and showed the appearance and disappearance of some regions but no global conformational changes. These modes were likely caused by differences in the strength of local densities among different particles and may not represent real gating-related conformational changes. The other mode exhibited global as well as local conformational changes including S6 dilation and contraction. This mode was then used to split the particles into 6 clusters using 3D-variability display (cluster mode), and a reconstruction was calculated for each cluster. Clusters with similar high-resolution features were combined. Two clusters corresponded to the closed state (74,921 particles). Two clusters corresponded to intermediate state 1 (131,492 particles). These 2 states were then subjected to a final round of non-uniform refinement. The density maps of the closed state and intermediate state 1 were refined to a resolution of 3.51 Å and 3.63 Å, respectively. The last two clusters (54,256 particles) with the most dilated S6 were subjected to another round of 3D-variability analysis,

with a hope to find an open-state class. Three modes were selected with a filter resolution of 4 Å. The resulting three modes were similar judging by 3D-variability display (simple mode). One of them was chosen to separate the particles into 4 clusters using 3D-variability display (cluster mode), and a reconstruction was calculated for each cluster. All clusters were similar and were combined as intermediate state 2 (54,256 particles) and subjected to a final round of non-uniform refinement. The density map of intermediate state 2 was refined to a resolution of 3.61 Å.

For the dataset of truncated CNGA3/CNGB3 in POPG/POPC nanodiscs, all steps were performed using cryoSPARC v3.3.2. 7689 movies were motion corrected and dose weighted using Patch Motion Correction job. Patch-based CTF estimation was performed on the aligned averages using Patch CTF. Blob Picker was used on the first 100 micrographs with 80–280 Å particle diameters. 126,621 particles were picked and extracted with 2 × binned (1.658 Å). Two rounds of 2D classification were performed and 40 good classes with different orientations were chosen as templates. Template Picker job was run on the full dataset of 7689 movies with particle diameter set to 175 Å. 5,265,916 particles were picked and extracted with 6 × binned (4.974 Å). Three rounds of 2D classifications were performed on the extracted particles. 1,179,242 particles were selected and extracted with 2 × binned (1.658 Å). Two rounds of Ab-initio reconstruction and heterogeneous refinement (3 classes) were performed to clean-up the particles in 3D. The selected 918,854 particles were used for consensus refinement (non-uniform refinement), resulting in a density map of 3.43 Å, which is at Nyquist resolution. It was confirmed that there was no misalignment of the subunits using the same method as described for the GDN dataset. The 3.43 Å map was used to generate a mask enclosing the whole CNGA3/CNGB3 channel without nanodisc density using UCSF Chimera[60]. Then, 3D-variability analysis was carried out using the generated mask, with three modes and a filter resolution of 3.5 Å. 3D-variability display (simple) was used to examine each mode. As in GDN, two of the three modes were similar and showed the appearance and disappearance of some regions but no global conformational changes. The other mode exhibited global as well as local conformational changes including S6 dilation and contraction. This mode was then used to split the particles into 5 clusters using 3D-variability display (cluster mode), and a reconstruction was calculated for each cluster. Clusters with similar high-resolution features were combined. Two clusters correspond to the closed state (336,268 particles). One cluster corresponded to intermediate state 1 (304,034 particles). One cluster corresponded to intermediate state 2 (208,556 particles). These 3 states were then subjected to a final round of non-uniform refinement. The density maps of the closed state, intermediate state 1 and intermediate state 2 were refined to a resolution of 3.11 Å, 3.33 Å, and 3.33 Å, respectively. The last cluster (68,636 particles) with the most dilated S6 was subjected to another round of 3D-variability analysis, with a hope to find an open-state class. Three modes were selected with a filter resolution of 3.5 Å. A mode with S6 rotation was identified by 3D-variability display (simple mode) and was chosen to separate the particles into 3 clusters using 3D-variability display (cluster mode) and a reconstruction was calculated for each cluster. Two clusters were similar and were combined (39,888 particles) and subjected to a final round of non-uniform refinement. The resulting density map was refined to a resolution of 3.33 Å and corresponded to the open state. The last cluster (28,535 particles) was subject to a 3rd round of 3D-variability analysis. Three modes were selected with a filter resolution of 3.7 Å. The resulting three modes were similar judging by 3D-variability display (simple mode). One of them was chosen to separate the particles into 5 clusters using 3D-variability display (cluster mode) and a reconstruction was calculated for each cluster. Three clusters yielded fragmented maps, and the corresponding particles were discarded. Two clusters were similar and were combined (15,177 particles) and subjected to a final round of non-uniform refinement. The resulting

density map was refined to a resolution of 3.60 Å and corresponded to a pre-open state.

3D-varibility display (intermediate mode) was used to generate the supplementary movies. For Supplementary Movie 1, 15 frames with 0 rolling window were generated using particles that are within 0.5% min/max range percentiles. A 3D map was generated for each frame and was filtered to 6 Å. For Supplementary Movie 2, 15 frames with 0 rolling window were generated using particles that are within 0.3% min/max range percentiles. A 3D map was generated for each frame and was filtered to 6 Å. For both movies, the first and last frames were discarded due to poor map resolution caused by a low particle number. Detergent micelle and nanodisc densities for each frame was removed. The final movies were prepared using ChimeraX and iMovie.

### Model building, refinement, and validation

Full-length apo CNGA3/CNBG3 structure (PDB ID: 7RHS) was used as the initial model. The initial model was docked into the cryo-EM maps using UCSF Chimera[60] and then refined against the cryo-EM map using phenix.real_space_refine[61] implemented in PHENIX[62] with the option simulated_annealing turned on. Afterward, model building was carried out in Coot[63] using the tool of Real-Space Refinement Zone by manually adding missed residues, remodeling unmatched regions and adjusting geometry and rotamer outliers. Coordinates and individual B-factors were finally refined against the cryo-EM map of heterotetrameric CNGA3/CNGB3 using phenix.real_space_refine in PHENIX with the option secondary structure restraints turned on and simulated_annealing off. FSC values were calculated between the map generated from the resulting model and the two half maps, as well as the averaged map of the two half maps. The quality of the models was evaluated with MolProbity[64]. All the figures were prepared in PyMol or UCSF ChimeraX[65]. Pore dimensions were analyzed using HOLE[66] program. CPOINT was set between the F392/F434 and V396/I438 cavity gate-forming residues. Detailed cryo-EM data collection, analysis, and refinement statistics are shown in Tables 1 and 2. Map resolution ranges were defined by the minimum and maximum values of local resolutions measured at all atom positions of each structure.

When comparing the structures of individual subunits in different states or conditions, we did not perform any alignments of PDBs to each other. Instead, for each cryo-EM dataset, the final refinement of the all the maps was performed using the same initial volume. This ensured that all maps were refined using the same spatial coordinates. Atomic models were built based on the refined maps and structural comparisons were made directly using the superimposed models without further alignments.

### Electrophysiology

Human embryonic kidney (HEK) 293T cells were grown in DMEM (Life Technologies 11965092) plus 10% newborn calf serum (Thermo 26010066) and penicillin (100 U/ml)/streptomycin (0.1 mg/ml) (Sigma P43333). LipoD293™ (SigmaGen Laboratories SL100668) was used for transfection. For full-length CNGA3/CNGB3, HEK 293T cells were co-transfected with pEZT-BM-MBP-CNGA3, pEZT-BM-Strep-CNGB3, and pEGFP-N1 with a ratio of 1:5:1. For truncated CNGA3/CNGB3, HEK 293T cells were co-transfected with pEZT-BM containing trunCNGA3-P2A-trunCNGB3 and pEGFP-N1 with a ratio of 5:1. Cells were split 5–7 h after transfection and used for recording 24–48 h after transfection.

All experiments were performed at room temperature (-22–23 °C). Pipettes were fabricated from borosilicate glass (World Precision Instruments) using a micropipette puller (P-1000, Sutter Instrument), and were fire-polished to resistances of 1–2 MΩ for inside-out patch recording. Both intracellular and extracellular solutions contained (in mM) 140 NaCl, 5 EGTA, and 10 HEPES (pH 7.4 adjusted with NaOH). Giga-Ω seal was formed by gentle suction when the patch pipette made contact with the cell. To obtain an inside-out membrane patch, the pipette was pulled away from the cell.

When necessary, the tip was exposed to air for <1 s and put back into the bath solution. Various concentrations of cGMP were added to the bath solution and were applied to the patch using a perfusion system. Currents were elicited by 100-ms voltage steps from −120 to +120 mV with 10-mV increments, with a holding potential of 0 mV. Currents were amplified by Axopatch 200B and digitized by Digidata 1440 A (Molecular Devices). Currents were low-pass filtered at 1 kHz and sampled at 10 kHz. pCLAMP 8.2 software (Molecular Devices), Microsoft Excel, and Prism 7 were used for data acquisition and analysis.

### MS-based quantitative lipidomics

CNGA3/B3 protein were purified as described before in 0.02% GDN. Protein sample was concentrated to 50 μL with a concentration of 6 mg/mL. Lipidomics analysis was performed by BGI San Joes Mass Spectrometry Center and the following experimental procedures were provided:

50 μL of the sample was added with 1 mL of extraction solution (MTBE: Methanol = 3:1, V/V) containing internal standards. The mixture was vortexed for 15 min. Next, the mixture was added with 200 μL of water, vortexed for 1 min, placed at 4 °C for 10 min, and centrifuged at 12,000 rpm for 10 min (4 °C). A 200 μL of the upper phase was transferred for complete solvent drying under 20 °C. The residue in the tube was added with 200 μL of reconstitution solution (ACN: IPA = 1:1, V/V), vortexed for 3 min, and centrifuged at 12,000 rpm for 3 min. A 120 μL of the final supernatant was transferred for LC-MS/MS analysis. Chromatography-mass spectrometry acquisition was conducted using Ultra Performance Liquid Chromatography (UPLC) (Nexera LC-40, https://www.shimadzu.com) and tandem mass spectrometry (MS/MS) (Triple Quad 6500+, https://sciex.com/).

### Liquid phase conditions.
(1)  Chromatographic column: Thermo AccucoreTMC30 (2.6 μm, 2.1 mm × 100 mm i.d.);
(2)  Mobile phase: A phase was acetonitrile/water (60/40, V/V) (0.1% formic acid added, 10 mmol/Lammonium formate); B phase was acetonitrile/Isopropyl alcohol (10/90, V/V) (0.1% formic acid added, 10 mmol/L ammonium formate);
(3)  Gradient program: 80:20(V/V) at 0 min, 70:30(V/V) at 2 min, 40:60(V/V) at 4 min, 15:85(V/V) at 9 min, 10:90(V/V) at 14 min, 5:95(V/V) at 15.5 min, 5:95(V/V) at 17.3 min, 80:20(V/V) at 17.5 min, 80:20(V/V) at 20 min; 4) Flow rate: 0.35 ml/min; Column temperature: 45 °C; Injection volume: 2 μL.

**Mass spectrometry conditions.** LIT and triple quadrupole (QQQ) scans were acquired on a triple quadrupole-linear ion trap mass spectrometer (QTRAP), QTRAP® 6500 + LC-MS/MS System, equipped with an ESI Turbo Ion-Spray interface, operating in positive and negative ion mode and controlled by Analyst 1.6.3 software (Sciex). The ESI source operation parameters were as follows: ion source, turbo spray; source temperature 500 °C; ion spray voltage (IS) 5500 V(Positive), −4500 V(Neagtive); Ion source gas 1 (GS1), gas 2 (GS2), curtain gas (CUR) were set at 45, 55, and 35 psi, respectively. Instrument tuning and mass calibration were performed with 10 and 100 μmol/L polypropylene glycol solutions in QQQ and LIT modes, respectively. QQQ scans were acquired as MRM experiments with collision gas (nitrogen) set to 5 psi. DP and CE for individual MRM transitions was done with further DP and CE optimization. A specific set of MRM transitions were monitored for each period according to the lipids eluted within this period.

**Lipid quantification.** Lipids were identified qualitatively based on its retention time and ion-pair information from MRM mode. In MRM mode, the first quadrupole screens the precursor ions for target substance and excluded ions of other molecular weights. After ionization induced by the impact chamber, the precursor ions were fragmented and a characteristic fragment ion was selected through the third

quadrupole to exclude the interference of non-target ions. By selecting a particular fragment, quantification is more accurate and consistent. Lipid concentration is calculated by the formula (nmol/mL): $X = 0.001*R*c*F*V / m$. X: Content of lipids in the sample (nmol/mL); R: The ratio of the peak area of the substance to be measured to the peak area of the internal standard (Area Ratio); V: Extraction solution for samples (μL); F: Internal standard correction factors for different types of substances; c: Concentration of internal (μmol/L); m: The sample size taken (mL), in this case, m = 0.05.

### Reporting summary

Further information on research design is available in the Nature Portfolio Reporting Summary linked to this article.

## Data availability

The authors declare that the data supporting the findings of this study are available within the paper. Full-length sequences of CNGA3 and CNGB3 are available at National Center for Biotechnology Information (NCBI) with reference code NP_001289.1 and NP_061971.3, respectively. The cryo-EM density maps generated in this study have been deposited in the Electron Microscopy Data Bank (https://www.ebi.ac.uk/pdbe/emdb/). For cGMP-bound full-length CNGA3/CNGB3 in GDN, the accession numbers for closed state, intermediate state 1, and intermediate state 2 are EMD-28595, EMD-28603, and EMD-28611, respectively. For cGMP-bound truncated CNGA3/CNGB3 in POPG/POPC nanodiscs, the accession numbers for closed state, intermediate state 1, intermediate state 2, pre-open state, and open state are EMD-28622, EMD-28623, EMD-28624, EMD-28625, and EMD-28626, respectively. The coordinates of the atomic models generated by this study have been deposited in the Protein Data Bank (http://www.rcsb.org). For cGMP-bound full-length CNGA3/CNGB3 in GDN, the accession numbers for closed state, intermediate state 1, and intermediate state 2 are 8ETP, 8EU3, and 8EUC, respectively. For cGMP-bound truncated CNGA3/CNGB3 in POPG/POPC nanodiscs, the accession numbers for closed state, intermediate state 1, intermediate state 2, pre-open state, and open state are 8EV8, 8EV9, 8EVA, 8EVB, and 8EVC, respectively. The atomic model for apo closed state CNGA3/CNGB3 in GDN 7RHS and corresponding cryo-EM map EMD-24468 are used for comparison and analysis. The mass spec data, electrophysiological data, and unprocessed SDS-PAGE images are available as Source data file online. Source data are provided with this paper.

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

## Acknowledgements

This research was supported by the National Institutes of Health grants RO1EY032880 and RO1GM085234 (J.Y.). Some of this work was performed at the Columbia University Cryo-Electron Microscopy Center (CUCEC) and at the Simons Electron Microscopy Center and National Resource for Automated Molecular Microscopy located at NYSBC, supported by grants from the Simons Foundation (SF349247), NYSTAR, and the NIH National Institute of General Medical Sciences (GM103310) with additional support from Agouron Institute (F00316) and NIH (OD019994). Some cryo-EM work was performed at the National Center for CryoEM Access and Training (NCCAT), supported by the NIH Common Fund Transformative High Resolution Cryo-Electron Microscopy program (U24 GM129539). We thank members of CUCEC, NYSBC, and NCCAT for support and assistance in cryo-EM grid screening and data acquisition. We thank members of our laboratory for discussion during the course of this work.

## Author contributions

Z.H. designed and performed molecular biological, biochemical, cryo-EM and electrophysiology experiments, built the atomic models, and analyzed results. X.Z. performed initial biochemical and cryo-EM experiments and assisted in atomic model building. J.Y. supervised the project, analyzed results, and wrote the paper with Z.H. and X.Z.

## Competing interests

The authors declare no competing interests.
