## [Peer Review File · Nature Communications]

Conformational trajectory of allosteric gating of the human cone photoreceptor cyclic nucleotide-gated channelREVIEWER COMMENTS

Reviewer #1 (Remarks to the Author):

The authors of this study provided in the past years excellent contributions to the field of CNG channels, with structures of TAX4-A (Li et al., Nature 2017) (Zheng et al., NSMB 2020), and the structure of the cone CNG channel in the closed state (Zhen et al., NSMB 2022). Here, the manuscript by Hu et al., describes the cryo-EM structures of the cone CNG channel in closed, transition, pre-open and open states. The authors provide a total of eight structures, three in detergent GDN, and five in lipid nanodiscs. The study is well conducted and timely, because structures of heteromeric CNG channels in the (full) open state have been long awaited. The authors had to circumvent a number of difficulties to obtain the pre-open and open state structures, which they finally achieved by improving sample quality, by using lipid nanodiscs, and by exploring the use of 3D-variability analysis and clustering to separate particles with subtle structural changes. The manuscript is well written and provides an important picture of the activation pathway of the cone CNG channel. The methodology described within the single-particle processing pipeline will be useful for the cryo-EM community to re-analyze datasets where ion channels were closed but bound to the ligand.

I have a few minor points that I wish the authors to consider during revision of this article.

Fig. 1E. The authors show that residue F392 on the A3L subunit adopts an outward conformation, different from the orientation towards the pore that F392 occupies in the other three subunits. The authors have previously reported (Zheng et al., NSMB 2022) that F392 can adopt two orientations (as they also describe in the text), a sideways projection, and a pore projection. Can the authors comment whether occupancy changes in the structures they report? One would expect that the occupancy for the pore projection would diminish towards the open state.

Fig. 1D. Was there a reason not to display one S6 segment from A3, and the S6 segment of B3, instead of two S6 segments from A3? This would allow to show also where R442 is located, and how it is oriented in the tree structures, which is an important link to Fig. 1F.

Fig. 1F. The pore profiles show that the largest difference between the three structures is near R442, but differently from the residues that compose the hydrophobic gate, in Fig. 1 there is no illustration of how R442 adopts a different orientation. It would be great if the authors could make an additional panel. If possible, density for R442 in the different structures could be shown as an additional supplementary figure.

I find Suppl. Fig. 9 rather important, but I have not seen it particularly discussed in the text. It would as well be useful to show how the interaction network of R442 changes between the closed and open structures in nanodiscs, because it was suggested that R442 in the human rod channel interacts with residues that are highly conserved on the S6 segment of the CNGB1 subunit (see Fig. 5B in "The structure of CNG channels in rod and cones photoreceptors" Trends in Neuroscience, 2022). For instance, in the open state, is R442 at contact distance to CNGA3 N398 or T394, or to a different residue?

The authors suggest that removing lipids from HEK cells that are co-purified with the channel is important to obtain a (small) fraction of the channel in pre-open (1.7%) and open states (4.3%). One important aspect that maybe would be worth discussing in more details is whether the channel actually needs these lipids in the native membrane to stabilize the closed state, and why stabilizing the closed state could be relevant for the function of cones and rods. Can the authors comment on the quality of the density maps in the closed and open conformation in nanodiscs, at the TM region, and whether this could affect recognizing lipids in the two maps, as those show the most important differences when it comes to lipids.

About sample preparation and constructs: the authors have EGTA present, which excludes the possibility to have the channel bound by calmodulin. Probably this should be clearly stated in the text. Is there any CaM binding site left in the nanodiscs construct?

The addition of cGMP at saturating concentrations (2-4 mM) might have a destabilizing effect on

the channel. Did the authors notice that the sample was more stable in nanodiscs after addition of cGMP, and was this resulting in a higher particle concentration on grids ?

The authors show that 3D-variability analysis and clustering is a good strategy to separate particles that belong to different conformational states, which is notoriously difficult to do by using conventional 3D classification methods, especially when structural changes are subtle, as in this case.

The authors should comment on whether the additional states identified by 3D-variability analysis and clustering could not be obtained in the GDN dataset only because the number of particles in this dataset was much lower than what obtained from the nanodiscs dataset (260,699 particles versus 918,854 particles). It is clear that with 260,699 particles it is more difficult to obtain a sufficient number of particles for a useful 3D reconstruction when the fraction of the open state is 4.3% (that would correspond to 10,427 particles out of 260,699). Would reprocessing of the GDN dataset entirely in cryoSPARC allow to retrieve more particles ?

In a recent paper (Zheng et al, Communications Biology 2022), the authors showed that different conformational states could be identified by using a mask that excluded part of the transmembrane region. It would be interesting if the authors could comment whether the same strategy would now be effective at separating the different states reported here.

The authors correctly state in the Discussion that experiments aimed at characterizing the nature of lipids that are co-purified with the channel are needed. These studies require careful planning and complementary techniques to cryo-EM, and should be considered for a separate study.

Reviewer #2 (Remarks to the Author):

The manuscript by Hu et al reports a series of structures solved by single-particle cryo-EM proposed to represent conformational intermediates in the cGMP activation pathway for the heteromeric CNG channel from human cone photoreceptors, composed of 3 alpha and 1 beta subunits. The structures were solved in both GDN detergent and 1:1 POPG/POPC lipid nanodiscs. They identified several intermediate and transition states, where the pre-open and open states are found only in the lipid nanodiscs, although the number of particles in these states constitute a small fraction of all the particles (1 and 4 %). The authors have two main take-home messages from these structures.

First, by examining the differences between the obtained structures, they propose a wave of conformational rearrangements that lead from cGMP binding to channel activation. One concern here is that the authors make subunit-specific claims, and the best approach to gain subunit specific information is symmetry expansion and classification on one subunit only, which the authors did not perform. The authors' approach, 3D variability can reveal global differences, but the different modes need to be explained in more detail. Nevertheless, this is a minor concern. The structures are impressive, this interpretation is solid, and fits the existing models for CNG channel gating.

Second, the authors analyze the non-protein densities present in these structure that can be reasonably attributed to lipids, assign them to specific lipids, and analyze their state-dependence. From this, they propose that certain lipids favor different channel conformations. There are multiple major concerns here. The identification of specific lipids (DAG, ceramide, etc...) based on only poor cryo-EM densities is not convincing. Most densities are not particularly well defined and some could very well be noise. Different filtering levels (sharpening b-factors) should be compared. Particle polishing in relion might help to improve the resolution of the structures in general, which also will dramatically help with the densities observed for lipids. Importantly, non-structure experiments (Mass spec, function, mutations) need to be performed to support such assignments of lipid identities. In addition, to support claims that certain lipids favor channel opening, it is not sufficient to compare "lipid densities" from different structures that are anyway at fairly low and different resolutions, functional experiments in these lipids are also required. In their absence, these are just hypotheses.

In summary, although the structures of the intermediate conformations of the cone CNG channels are compelling and likely define an allosteric trajectory along the activation of these channel by their endogenous ligand, the interpretations of the structures bound with lipids, the assignments of specific lipids into densities, and the extrapolations of these lipid assignments into functional phenotypes are rather unsupported and require either further experiments or removal from the manuscript.

Response to Reviewers

We thank the reviewers for their insightful and helpful questions and suggestions. We have revised the manuscript extensively based on their comments. In particular, we restructure the manuscript to focus predominantly on reporting and describing the structures and conformational changes of A3/B3 in different states and deemphasize presumed state-dependent lipid binding. We hope and believe that the manuscript is much improved as a result. Main changes include:

- (1). The entire last section of Results, titled “State-dependent lipid binding”, in the original manuscript is deleted.
- (2). The original Fig. 6 and Supplementary Fig. 10 are deleted.
- (3). Some original main and supplementary figures are reorganized to generate the current Figs. 3-6.
- (4). A new Fig. 7 and new Supplementary Figs. 3 and 4 are added.
- (5). The original Fig. 5 is relocated from Results to Discussion and is now Supplementary Fig. 10.
- (6). We performed mass spectroscopy of a purified A3/B3 protein sample reconstituted in GDN. The result shows that many different types of lipids are indeed present in the sample, including cholesterol, phosphatidylserine, phosphatidylethanolamine, phosphatidylcholine, ceramide and diacylglycerol. This unpublished result is mentioned in Discussion but is not presented because it is not reinforced by other functional and mutagenesis results.
- (7). Relevant texts are added or modified throughout the manuscript. They are highlighted in yellow.

Below are point-to-point responses to the reviewers. Reviewers’ comments are in *italic*.

Reviewer #1 (Remarks to the Author):

The authors of this study provided in the past years excellent contributions to the field of CNG channels, with structures of TAX4-A (Li et al, Nature 2017) (Zheng et al., NSMB 2020), and the structure of the cone CNG channel in the closed state (Zhen et al., NSMB 2022).

Here, the manuscript by Hu et al., describes the cryo-EM structures of the cone CNG channel in closed, transition, pre-open and open states. The authors provide a total of eight structures, three in detergent GDN, and five in lipid nanodiscs. The study is well conducted and timely, because structures of heteromeric CNG channels in the (full) open state have been long awaited.

The authors had to circumvent a number of difficulties to obtain the pre-open and open state structures, which they finally achieved by improving sample quality, by using lipid nanodiscs, and by exploring the use of 3D-variability analysis and clustering to separate particles with subtle structural changes. The manuscript is well written and provides an important picture of the activation pathway of the cone CNG channel. The methodology described within the single-particle processing pipeline will be useful for the cryo-EM community to re-analyze datasets where ion channels were closed but bound to the ligand.

ANSWER: We thank the reviewer for these positive and encouraging comments.

I have a few minor points that I wish the authors to consider during revision of this article.

Fig. 1E. The authors show that residue F392 on the A3L subunit adopts an outward conformation, different from the orientation towards the pore that F392 occupies in the other three subunits. The authors have previously reported (Zheng at al., NSMB 2022) that F392 can adopt two orientations (as they also describe in the text), a sideways projection, and a pore projection. Can the authors comments whether occupancy changes in the structures they report? One would expect that the occupancy for the pore projection would diminish towards the open state.

ANSWER: Yes, it does, and the reviewer's expectation is correct. In response we add a new Supplementary Figure 3 (showing local cryo-EM density maps of F392/F434 in all 8 reported structures) and add/modify the following texts:

Lines 137-141: "The other gate residue F392 of A3_L has two projections (sideway projection and pore projection) with similar occupancy in all three states, as in GDN_{apo}³⁴ (Supplementary Fig. 3a-c). In contrast, F392 of A3_D and A3_R and F434 of B3 have only pore projection (Supplementary Fig. 3a-c)."

Lines 272-276: "As in GDN, the gate residue F392 of A3_L has a sideways projection and a pore projection with similar occupancy in ND_{cGMP}_closed, ND_{cGMP}_transition1, ND_{cGMP}_transition2 and ND_{cGMP}_pre-open, whereas F392 of A3_D and A3_R and F434 of B3 have only pore projection in all these states (Supplementary Fig. 3d-g)."

Lines 286-289: "In ND_{cGMP}_open, the side-chains of all gate residues, including F392 in A3_L, project away from the pore, rendering the cavity gate fully open (Fig. 6d, g and Supplementary Fig. 3h)."

Fig. 1D. Was there a reason not to display one S6 segment from A3, and the S6 segment of B3, instead of two S6 segments from A3? This would allow to show also where R442 is located, and how it is oriented the tree structures, which is an important link to Fig. 1F.

ANSWER: The S6 segments displayed in the original Fig. 1D are A3 S6 segments with the most and the least conformational changes. Per reviewer's suggestion, we add the

S6 of B3 in Fig. 1D. The issue regarding R442 projections in different structures is addressed below.

Fig. 1F. The pore profiles show that the largest difference between the three structures is near R442, but differently from the residues that compose the hydrophobic gate, in Fig. 1 there is no illustration of how R442 adopts a different orientation. It would be great if the authors could make an additional panel. If possible, density for R442 in the different structures could be showed as an additional supplementary figure.

ANSWER: We thank the reviewer for these suggestions. In response, we add a new Supplementary Figure 4 (showing local cryo-EM density maps of R442 in all 3 structures in GDN and 5 structures in lipid nanodisc) and a new Fig. 7 (showing projections and interactions of R442), and add/modify the following texts:

Lines 141-147: “R442 of B3, which projects to the pore and forms the arginine gate below the cavity gate³⁴, also has two orientations (up and down) along the ion conduction pathway (Supplementary Fig. 4a-c). The up orientation is more abundant in GDN_cGMP_closed and the down orientation is more abundant in GDN_cGMP_transition1, while the density of R442’s distal side-chain is not well resolved in GDN_cGMP_transition2 (Supplementary Fig. 4a-c). These results suggest that the side-chains of F392 of A3_L and R442 of B3 exhibit some intrinsic flexibility.”

Lines 293-300: “The arginine gate formed by R442 of B3 also undergoes gradual conformational changes during closed-to-open transitions and opens fully only in the open state (Fig. 7). In ND_cGMP_closed and ND_cGMP_transition1, R442 projects to the ion conduction pathway with up and down orientations (Fig. 7a, b and Supplementary Fig. 4d, e), as in GDN. The up orientation diminishes in ND_cGMP_transition2 and ND_cGMP_pre-open (Fig. 7c, d and Supplementary Fig. 4f, g). In ND_cGMP_open, the R442 side-chain swings from the center of the pore to the side and opens the arginine gate (Fig. 7e, f and Supplementary Fig. 4h).”

I find Suppl. Fig. 9 rather important, but I have not seen it particularly discussed in the text. It would as well be useful to show how the interaction network of R442 changes between the closed and open structures in nanodiscs, because it was suggested that R442 in the human rod channel interacts with residues that are highly conserved on the S6 segment of the CNGB1 subunit (see Fig. 5B in “The structure of CNG channels in rod and cones photoreceptors” Trends in Neuroscience, 2022). For instance, in the open state, is R442 at contact distance to CNGA3 N398 or T394, or to a different residue?

ANSWER: We thank the reviewer for this suggestion and for giving us the opportunity to describe in detail state-dependent changes of R442 projections and interactions in POPG/POPC nanodisc. In response, we add a new Fig. 7 (showing projections and interactions of R442) and the following texts:

Lines 300-310: “R442 is engaged in different interactions with S6 of A3_L and A3_R in different states (Fig. 7a-e). In ND_cGMP_closed, R442 interacts with V396 of A3_R in the

up orientation but with S404 of A3_L in the down orientation (Fig. 7a). In ND_cGMP_transition1, the R442-V396 interaction is maintained but the R442-S404 interaction is lost (Fig. 7b). Both interactions are lost in ND_cGMP_transition2, ND_cGMP_pre-open and ND_cGMP_open (Fig. 7c-e), but in the open state R442 is engaged in a new interaction as a result of its rotation, forming a strong hydrogen bond with G397 of A3_L (Fig. 7e). Rotamer conformations and interactions with S6 of neighboring subunits have also been observed for the arginine gate in rod CNGA1/CNGB1 channels^{30,31,33,34}. The functional importance, if any, of a flexible arginine gate in cone and rod photoreceptor CNG channels remains to be elucidated³¹.”

The authors suggest that removing lipids from HEK cells that are co-purified with the channel is important to obtain a (small) fraction of the channel in pre-open (1.7%) and open states (4.3%). One important aspect that maybe would be worth discussing in more details is whether the channel actually needs these lipids in the native membrane to stabilize the closed state, and why stabilizing the close state could be relevant for the function of cones and rods.

ANSWER: The issues brought up by the reviewer are interesting and important. However, based on Reviewer 2's suggestion and with further consideration, we decide to remove most of the previous results, figures and discussion related to lipid binding to A3/B3 and its role in gating. The entire last section of Results, titled “State-dependent lipid binding”, in the original manuscript is deleted. The original Fig. 6 and Supplementary Fig. 10 are deleted, and the original Fig. 5 is moved from Results to Discussion and is now Supplementary Fig. 10. We leave only the last section in Discussion to discuss a key role of lipids in shaping the energetic landscape of cGMP activation of A3/B3 and what needs to be done in this direction (lines 358-384). We mention that “many different types native lipids are robustly detected by mass spectroscopy of a purified A3/B3 protein sample reconstituted in GDN, including cholesterol, phosphatidylserine, phosphatidylethanolamine, phosphatidylcholine, ceramide and diacylglycerol (unpublished data).” But we feel that in the absence of more definitive results (especially functional mutagenesis results), it is premature and too speculative to discuss the points raised by the reviewer. Hopefully future studies will provide answers to these points. We hope the reviewer understands and agrees with this sentiment.

Can the authors comment on the quality of the density maps in the closed and open conformation in nanodiscs, at the TM region, and whether this could affect recognizing lipids in the two maps, as those show the most important differences when it comes to lipids.

ANSWER: The resolution of transmembrane domains ranges from ~2.8 Å to ~3.4 Å in the closed and open states in nanodisc, but nonprotein densities tend to have lower resolutions, making it difficult to definitively model lipids into the densities. As Reviewer 2 points out, “non-structure experiments (Mass spec, function, mutations) need to be performed to support such assignments of lipid identities.” We agree with Reviewer 2 but think that these experiments will take a long time to complete and are more suitable

for a separate study; as such, we decide to remove the original Fig. 6 and Supplementary Fig. 10.

About sample preparation and constructs: the authors have EGTA present, which excludes the possibility to have the channel bound by calmodulin. Probably this should be clearly stated in the text. Is there any CaM binding site left in the nanodiscs construct?

ANSWER: The following sentence is added: "Calmodulin was unlikely bound to our A3/B3 sample because 2 mM EGTA was included during protein purification, and no density corresponding to calmodulin was observed in any density maps." (Lines 342-344)

A putative calmodulin binding site is still present in the truncated CNGB3 subunit.

The addition of cGMP at saturating concentrations (2-4 mM) might have a destabilizing effect on the channel. Did the authors notice that the sample was more stable in nanodiscs after addition of cGMP, and was this resulting in a higher particle concentration on grids?

ANSWER: A3/B3 proteins were uniform and stable before adding cGMP, judging from gel filtration curves and SDS-PAGE. We never observed a destabilizing effect of adding saturating concentrations of cGMP to either TAX-4 or CNGA3/CNGB3. The reason that the nanodisc sample had a higher particle density than the GDN sample did is because the nanodisc sample was concentrated to a higher concentration than the GDN sample (2 mg/ml vs 1.5 mg/ml) before grid making.

The authors show that 3D-variability analysis and clustering is a good strategy to separate particles that belong to different conformational states, which is notoriously difficult to do by using conventional 3D classification methods, especially when structural changes are subtle, as in this case.

The authors should comment on whether the additional states identified by 3D-variability analysis and clustering could not be obtained in the GDN dataset only because the number of particles in this dataset was much lower than what obtained from the nanodiscs dataset (260,699 particles versus 918,854 particles). It is clear that with 260,699 particles it is more difficult to obtain a sufficient number of particles for a useful 3D reconstruction when the fraction of the open state is 4.3% (that would correspond to 10,427 particles out of 260,699). Would reprocessing of the GDN dataset entirely in cryoSPARC allow to retrieve more particles?

ANSWER: The reviewer brings up a good question. To investigate whether the larger particle number is the reason we obtained more states in the nanodisc sample, we randomly selected a subset of nanodisc particles and performed the same analysis. We used 250,000 particles of the nanodisc dataset, which is less than the particles used for the GDN sample (260,669). After 3D-variability analysis, we can still obtain the fully open state with 9661 particles (3.8%) with 3.86 Å resolution. Although the resolution is

not as high, we can still resolve side-chain orientations of most transmembrane residues and cavity gate is fully open.

With regard to reviewer's second question, we believe that it is possible to retrieve more particles if we redo the analysis entirely in cryosparc, especially with topaz training and extracting. However, based on the reprocessing result mentioned above, we believe that the number of particles is not a major reason why the fully open state is absent in the GDN sample.

In a recent paper (Zheng et al, Communications Biology 2022), the authors showed that different conformational states could be identified by using a mask that excluded part of the transmembrane region. It would be interesting if the authors could comment whether the same strategy would now be effective at separating the different states reported here.

ANSWER: We thank the reviewer for this suggestion. We reprocessed the dataset from the nanodisc sample, using the same strategy as in our recent TAX-4 paper. We put a mask on S6, the gating ring and the CNBD, just as the TAX-4 paper described. We then performed both focused 3D classification in Relion and 3D-variability analysis in cryosparc. However, the result was not nearly as good as our current strategy. We were able to see some conformational changes between different states, but was unable to obtain the pre-open and open states. The reason we masked only S6, the gating ring and the CNBD in the TAX-4 paper was that these regions underwent the largest conformational changes from closed to open. We now find that it is also important to include regions with less conformational changes in the mask.

The authors correctly state in the Discussion that experiments aimed at characterizing the nature of lipids that are co-purified with the channel are needed. These studies require careful planning and complementary techniques to cryo-EM, and should be considered for a separate study.

ANSWER: We thank the reviewer for this understanding.

Reviewer #2 (Remarks to the Author):

The manuscript by Hu et al reports a series of structures solved by single-particle cryo-EM proposed to represent conformational intermediates in the cGMP activation pathway for the heteromeric CNG channel from human cone photoreceptors, composed of 3 alpha and 1 beta subunits. The structures were solved in both GDN detergent and 1:1 POPG/POPC lipid nanodiscs. They identified several intermediate and transition states, where the pre-open and open states are found only in the lipid nanodiscs, although the number of particles in these states constitute a small fraction of all the particles (1 and 4 %). The authors have two main take-home messages from these structures.

First, by examining the differences between the obtained structures, they propose a wave of conformational rearrangements that lead from cGMP binding to channel

activation. One concern here is that the authors make subunit-specific claims, and the best approach to gain subunit specific information is symmetry expansion and classification on one subunit only, which the authors did not perform. The authors' approach, 3D variability can reveal global differences, but the different modes need to be explained in more detail. Nevertheless, this is a minor concern. The structures are impressive, this interpretation is solid, and fits the existing models for CNG channel gating.

ANSWER: We thank the reviewer for this suggestion. We think symmetry expansion and classification on one subunit may not be suitable in the case of A3/B3. This heteromeric channel has a 3 A3: 1 B3 stoichiometry and thus has no symmetry. Each subunit interacts with its neighboring subunits through different and unique interfaces. Given this, one would expect that there is only one correct way for the four subunits and hence the particles to align. We therefore used C1 symmetry throughout data processing. In response to the reviewer's question, and to check if it is possible that some of the particles were misaligned (e.g., A3_L of some particles were aligned to A3_D or A3_R of the consensus map), we performed focused 3D classification focusing on the B3 subunit of the consensus map. Our reasoning was that if there was a misalignment of the particles, the B3 subunit of the consensus map would be a mixture of A3 and B3 subunits. The result showed that the B3 subunit of the consensus map contained only B3. This B3-focused analysis is now included in Imaging Processing in Methods (lines 526-535 and 567-568). Based on these considerations and results, we are confident that there is no misalignment of the particles and the subunit-specific conformational changes are real and are not averages of multiple misaligned subunits.

In addition to the S6 dilation-contraction mode described in the paper, two other similar modes are observed, which mainly involve the appearance and disappearance of some regions. These modes may be caused by differences in the local density strength among different particles and may not represent real gating-related conformational changes. The three modes are now described in more detail in Methods (lines 537-542 and 577-580).

Second, the authors analyze the non-protein densities present in these structure that can be reasonably attributed to lipids, assign them to specific lipids, and analyze their state-dependence. From this, they propose that certain lipids favor different channel conformations. There are multiple major concerns here. The identification of specific lipids (DAG, ceramide, etc...) based on only poor cryo-EM densities is not convincing. Most densities are not particularly well defined and some could very well be noise. Different filtering levels (sharpening b-factors) should be compared. Particle polishing in relion might help to improve the resolution of the structures in general, which also will dramatically help with the densities observed for lipids. Importantly, non-structure experiments (Mass spec, function, mutations) need to be performed to support such assignments of lipid identities. In addition, to support claims that certain lipids favor channel opening, it is not sufficient to compare "lipid densities" from different structures that are anyway at fairly low and different resolutions, functional experiments in these lipids are also required. In their absence, these are just hypotheses.

In summary, although the structures of the intermediate conformations of the cone CNG channels are compelling and likely define an allosteric trajectory along the activation of these channel by their endogenous ligand, the interpretations of the structures bound with lipids, the assignments of specific lipids into densities, and the extrapolations of these lipid assignments into functional phenotypes are rather unsupported and require either further experiments or removal from the manuscript.

ANSWER: We thank the reviewer for this constructive critique and suggestion. We understand and appreciate the reviewer's concerns, and we agree with the reviewer's assessment. The experiments mentioned by the reviewer will indeed be necessary and informative in elucidating the role and mechanism of lipid modulation of A3/B3 gating. These experiments, however, will require careful planning and design and will likely take a long time to complete. We feel they are more suitable for a separate study (as echoed by Reviewer 1). Thus, as the reviewer suggested, we decide to remove most of the previous results, figures and discussion related to lipid binding to A3/B3 and its role in gating: (i) the original sentence "Different states also display different binding profiles of endogenous and exogenous lipids" in the Abstract is deleted; (ii) the entire last section of Results, titled "State-dependent lipid binding", in the original manuscript is deleted; (iii) the original Fig. 6 and Supplementary Fig. 10 are deleted; (iv) the original Fig. 5 is moved from Results to Discussion and is now Supplementary Fig. 10. We leave only the last section in Discussion to discuss a key role of lipids in shaping the energetic landscape of cGMP activation of A3/B3 and what needs to be done in this direction (lines 358-384).

As mentioned at early in the response, we performed mass spectroscopy of a purified A3/B3 protein sample reconstituted in GDN. The result shows that many different types of lipids are indeed present in the sample, including cholesterol, phosphatidylserine, phosphatidylethanolamine, phosphatidylcholine, ceramide and diacylglycerol. This unpublished result is mentioned in Discussion but is not presented because it is not reinforced by other functional and mutagenesis results.

With the deletions of lipid-related figures and texts, and in response to reviewers' comments, we significantly expand descriptions of the conformational changes of key regions and amino acids in the revised manuscript.

REVIEWERS' COMMENTS

Reviewer #1 (Remarks to the Author):

The authors have addressed all my concerns and suggestions in a very satisfactory manner. The manuscript has undergone substantial changes and as consequence has improved significantly.

The manuscript is solid in its current form, which benefits both the authors and readers. I encourage the authors to continue their work on the role of lipids in the opening mechanism of CNG channels. It will be a game-changer in the field.

The authors have performed additional processing of their cryo-EM datasets to fully support their (already excellent) processing strategy. This is a time-consuming process and I thank the authors for their efforts. I am glad to see that densities for R442 have been reported. Figure 5 is important and will be a reference for future structures. Finally, I find Figure 8 useful, but I would suggest the authors to go further, label the A and B subunits, and add the arginine gate, which is currently absent. I think the authors have sufficient information on the movement of S6 and R442 of CNGB3 to add the arginine gate to the model. If space is limiting, the authors could consider organizing the figure on two layers or as a circle.

I recommend this manuscript to be published without delays.

Reviewer #2 (Remarks to the Author):

The revised manuscript no longer contains the analysis of the lipids. However, the authors may have gone too far with this. My concern was only related to modeling specific lipids into specific densities, I did not question that lipids are bound to these channels. I also think the authors should include the mass spectrometry analysis they allude to in the discussion but state as "not published". This is a result, and it should be in the paper, as long as it is stated as follows or similar: "these are the lipids we identified in the sample.". It is in fact quite interesting that they find all these different lipids in the sample.

The figures have been restructured to include mostly information about the conformational changes occurring in the different subunits during opening, going through the identified intermediates. Some of the figures, such as Fig 3, 4, and 6 are meant to illustrate the gradual changes in the different domains: CNBD-C-linker, TMs 4-6, and cavity gate, respectively. I don't find these figures particularly compelling to illustrate the authors' important findings because they show each stepwise change in a series of overlays with teeny tiny changes and the effect is underwhelming. 3 figures for this is excessive and can easily be replaced by one figure with 3 panels showing one overlay of all the conformations for each domain with some smart coloring and perspective.

The text discusses at length side chain re-arrangements. However, cryo-EM density is missing from the main figures. Side chain movements, when central to the description of conformational changes, should be shown with the respective densities.

Otherwise, this is beautiful work, amazing structures and findings.

Other comments:

L 132-133: "The magnitude of the local conformational changes varies from subunit to subunit and is larger in GDN_cGMP_transition2 than in GDN_cGMP_transition1 (Fig1c-e)."

By comparing individual subunits in heteromeric channels the authors state subunit specific changes (based on their PDBs shown in Fig 1). It would be useful to indicate how the structures were aligned. Simply aligning PDBs to each other can place excessive weight on the A subunit, which can artificially displace the other subunits to different extents. The outcome could be that the gradually increasing changes they describe originate from poorly aligned PDBs.

L212-215: "The distributions of the different states in GDN and POPG/POPC nanodiscs indicate that copurified native lipids, not GDN, arrest A3/B3 in the closed or transition states and POPG/POPC

can liberate some channels from these states and enable them to open."

It is unclear why native lipids would be inhibiting. How can the authors differentiate between their scenario and the possibility that important lipids are missing in the detergent structure? The results can be more straightforwardly interpreted as "in nanodiscs the more native-like, lipid environment enables channel opening"

L224-225: "pair since they exhibit the most prominent structural changes"

Same as first comment. It is important to know how the structures were aligned.

L246: "These movements are concrete, in sequence, and in the expected direction"

"concrete, in sequence" are unclear.

L261: "strength of gating ring-TMD interactions increase gradually"

It is difficult to discuss strength of interactions from just structures. At the least, the authors should include the distances of the interactions in the Figure (Fig 5) to allow the reader to assess these statements.

L264: "with more and stronger interactions in the open state such graded interactions"

Same as previous comment. At least distances need to be provided.

L286: "albeit its cavity gate may become somewhat leaky"

Given the "maybe" and "somewhat", it is difficult to understand why this sentence is here.

Response to Reviewers

We thank the reviewers for their second round comments. We have revised the manuscript based on their suggestions, which help improve the manuscript further. Below are point-to-point responses to the reviewers. Reviewers' comments are in italic.

Reviewer #1 (Remarks to the Author):

The authors have addressed all my concerns and suggestions in a very satisfactory manner. The manuscript has undergone substantial changes and as consequence has improved significantly.

The manuscript is solid in its current form, which benefits both the authors and readers. I encourage the authors to continue their work on the role of lipids in the opening mechanism of CNG channels. It will be a game-changer in the field.

The authors have performed additional processing of their cryo-EM datasets to fully support their (already excellent) processing strategy. This is a time-consuming process and I thank the authors for their efforts. I am glad to see that densities for R442 have been reported. Figure 5 is important and will be a reference for future structures. Finally, I find Figure 8 useful, but I would suggest the authors to go further, label the A and B subunits, and add the arginine gate, which is currently absent. I think the authors have sufficient information on the movement of S6 and R442 of CNGB3 to add the arginine gate to the model. If space is limiting, the authors could consider organizing the figure on two layers or as a circle.

I recommend this manuscript to be published without delays.

ANSWER: We thank the reviewer for the positive and encouraging comments. As suggested by the reviewer, we color-code and label the A3 and B3 subunits and add the arginine gate in Fig. 8.

Reviewer #2 (Remarks to the Author):

The revised manuscript no longer contains the analysis of the lipids. However, the authors may have gone too far with this. My concern was only related to modeling specific lipids into specific densities, I did not question that lipids are bound to these channels. I also think the authors should include the mass spectrometry analysis they allude to in the discussion but state as “not published”. This is a result, and it should be in the paper, as long as it is stated as follows or similar: “these are the lipids we identified in the sample.”. It is in fact quite interesting that they find all these different lipids in the sample.

ANSWER: We now have a better understanding of the reviewer's previous concerns, and based on the reviewer's new comments and suggestions, we add a section titled "Lipid binding" to Results. This short section describes the existence of abundant non-protein densities in purified A3/B3 samples and the identification by mass spectroscopy of many different types of lipids in a purified A3/B3 protein sample reconstituted in GDN.

The figures have been restructured to include mostly information about the conformational changes occurring in the different subunits during opening, going through the identified intermediates. Some of the figures, such as Fig 3, 4, and 6 are meant to illustrate the gradual changes in the different domains: CNBD-C-linker, TMs 4-6, and cavity gate, respectively. I don't find these figures particularly compelling to illustrate the authors' important findings because they show each stepwise change in a series of overlays with teeny tiny changes and the effect is underwhelming. 3 figures for this is excessive and can easily be replaced by one figure with 3 panels showing one overlay of all the conformations for each domain with some smart coloring and perspective.

ANSWER: We appreciate the reviewer's concern and perspective about this presentation issue. In response, we combine previous figures 3 and 4 into a new Fig. 3. In this new figure, overlays of S4-S5 and S6 in all five structures are shown in Fig. 3c and 3d, respectively. We keep the pairwise comparisons in Fig. 3b because we think these comparisons reveal important information on the incremental conformational changes taken place in the CNBD and C-linker/gating ring. An overlay of these regions in all five structures can be seen in Fig. 2d and is insufficient to convey the messages demonstrated by the pairwise comparisons.

We also leave the previous Fig. 6 (now Fig. 5) unchanged. This figure describes the gradual conformational changes in S6 and the cavity-gate forming residues. We feel that only pairwise comparisons are clear enough to illustrate these conformational changes.

The text discusses at length side chain re-arrangements. However, cryo-EM density is missing from the main figures. Side chain movements, when central to the description of conformational changes, should be shown with the respective densities.

ANSWER: This is a good point, and we generally agree with the reviewer. In response, in Fig. 6 we include the densities of R442 in all five states. However, it is difficult to include the densities of all the residues that form the cavity gate in the main figures because there are simply too many densities. As such, they are shown only in supplemental figures (Supplementary Figs. 3 and 4).

Otherwise, this is beautiful work, amazing structures and findings.

ANSWER: We thank the reviewer for this positive and encouraging comment.

Other comments:

L 132-133: "The magnitude of the local conformational changes varies from subunit to

subunit and is larger in GDN_cGMP_transition2 than in GDN_cGMP_transition1 (Fig1c-e)."

By comparing individual subunits in heteromeric channels the authors state subunit specific changes (based on their PDBs shown in Fig 1). It would be useful to indicate how the structures were aligned. Simply aligning PDBs to each other can place excessive weight on the A subunit, which can artificially displace the other subunits to different extents. The outcome could be that the gradually increasing changes they describe originate from poorly aligned PDBs.

ANSWER: The reviewer brings up a valid concern. We did not perform any alignments of PDBs to each other when we compared the structures of individual subunits in different states/conditions. For each cryo-EM-dataset, the final refinement of the all the maps was performed using the same initial volume. This ensured that all maps were refined using the same spatial coordinates. Atomic models were built based on the refined maps and structural comparisons were made directly using the superimposed models without further alignments. These details about structural comparisons are now included in Methods.

L212-215: "The distributions of the different states in GDN and POPG/POPC nanodiscs indicate that copurified native lipids, not GDN, arrest A3/B3 in the closed or transition states and POPG/POPC can liberate some channels from these states and enable them to open."

It is unclear why native lipids would be inhibiting. How can the authors differentiate between their scenario and the possibility that important lipids are missing in the detergent structure? The results can be more straightforwardly interpreted as "in nanodiscs the more native-like, lipid environment enables channel opening"

ANSWER: In the first version of the manuscript, we mentioned the well-documented role of DAG in inhibiting CNG channel activity. This discussion was deleted in the last revision when we removed most of the results, figures and discussion related to lipid binding to A3/B3 and its role in modulating gating. In light of our new data on the identification of DAG in our purified A3/B3 protein sample in GDN, we postulate in Discussion that A3/B3 is inhibited by copurified DAG molecules and that POPG/POPC can displace some bound DAG.

*L224-225: "pair since they exhibit the most prominent structural changes"
Same as first comment. It is important to know how the structures were aligned.*

ANSWER: See the answer to the first comment in "Other comments".

*L246: "These movements are concrete, in sequence, and in the expected direction"
"concrete, in sequence" are unclear.*

ANSWER: These words are deleted.

*L261: "strength of gating ring-TMD interactions increase gradually"
It is difficult to discuss strength of interactions from just structures. At the least, the*

authors should include the distances of the interactions in the Figure (Fig 5) to allow the reader to assess these statements.

*L264: "with more and stronger interactions in the open state such graded interactions"
Same as previous comment. At least distances need to be provided.*

ANSWER: Distances of the interactions are indicated in the current Fig. 4 (previous Fig. 5), as suggested by the reviewer.

*L286: "albeit its cavity gate may become somewhat leaky"
Given the "maybe" and "somewhat", it is difficult to understand why this sentence is here.*

ANSWER: This sentence is deleted.